# Impacts of atmospheric circulation patterns and cloud inhibition on aerosol radiative effect and boundary layer structure during winter air pollution in Sichuan Basin, China

**Hua Lu[1,3], Min Xie[2,6], Bingliang Zhuang[1], Bojun Liu[4], Yangzhihao Zhan[1], Tijian Wang[1], Shu Li[1], Mengmeng Li[1], Kuanguang Zhu[1,5]**

[1]School of Atmospheric Sciences, Nanjing University, Nanjing 210023, China

[2]School of Environment, Nanjing Normal University, Nanjing 210023, China

[3] Chongqing Institute of Meteorological Sciences, Chongqing 401147, China

[4]Chongqing Meteorological Observatory, Chongqing 401147, China

[5]Hubei Provincial Academy of Eco-environmental Sciences, Wuhan 430079, China

[6]Carbon monitoring and digital application technology center, Carbon peak and carbon neutralization strategy institute of Jiangsu Province, Nanjing 210023, China

# Impacts of atmospheric circulation patterns and cloud inhibition on aerosol radiative effect and boundary layer structure during winter air pollution in Sichuan Basin, China

Hua Lu[1,3], Min Xie[2,6], Bingliang Zhuang[1], Bojun Liu[4], Yangzhihao Zhan[1], Tijian Wang[1], Shu Li[1], Mengmeng Li[1], Kuanguang Zhu[1,5]

[1]School of Atmospheric Sciences, Nanjing University, Nanjing 210023, China

[2]School of Environment, Nanjing Normal University, Nanjing 210023, China

[3] Chongqing Institute of Meteorological Sciences, Chongqing 401147, China

[4]Chongqing Meteorological Observatory, Chongqing 401147, China

[5]Hubei Provincial Academy of Eco-environmental Sciences, Wuhan 430079, China

[6]Carbon monitoring and digital application technology center, Carbon peak and carbon neutralization strategy institute of Jiangsu Province, Nanjing 210023, China

*Correspondence to*: Min Xie (minxie@njnu.edu.cn)

**Abstract.** Winter persistent aerosol pollution frequently occurs in the Sichuan Basin (SCB) due to its unfavorable weather conditions, such as low wind, wetness, and cloudiness. Based on long – term observational data analyses from 2015–2021, it has been found that the four representative stations in the SCB often simultaneously experience $PM_{2.5}$ pollution accompanied by variations in meteorological conditions above 850 hPa, which indicates a connection between regional winter air pollution in the SCB and large–scale synoptic patterns. The dominant 850 hPa synoptic patterns of winter SCB were classified into six patterns using T–model principal component analysis: (1) strong high pressure in the north, (2) east high west low (EHWL) pressure, (3) weak high pressure in the north, (4) weak ridge of high pressure after the trough, (5) low trough (LT), and (6) strong high pressure. Pattern 2 characterized with EHWL pressure system, and Pattern 5 featured with LT, were identified as key synoptic patterns for the beginning and accumulation of pollution processes. Pattern 1, characterized by a strong high pressure in the north, was the cleanest pattern associated with reduced $PM_{2.5}$ concentrations. The EHWL and LT patterns were associated with a remarkably high cloud liquid content, attributed to upper southerly winds introducing humid air. Clouds reduce solar radiation through reflection and scattering, resulting in more stable stratification and aerosol accumulation. This cloud radiation interaction (CRI) was more pronounced in the LT pattern due to denser isobaric lines and stronger southerly winds than in the EHWL pattern. Numerical simulation experiments utilizing WRF-Chem indicated that there is a upper-level heating during afternoon and surface cooling in the morning forced by the aerosol radiation interaction (ARI) under the EHWL and LT patterns. Additionally, strong surface cooling in the evening influenced by valley winds could be found. With wet and cloudy synoptic forcing, ARI directly affects the stability of the boundary layer and is modulated through CRI inhibition. For example, Chongqing exhibited lower $PM_{2.5}$ concentrations and stronger ARI compared to the western and southern SCB due to lower cloud liquid content and weaker CRI inhibition on ARI. The CRI inhibition caused a 50 % reduction in solar radiation and boundary

layer height during the daytime under the LT pattern, which was larger than that under the EHWL
pattern. This study comprehensively analyzed the spatial disparities in cloud inhibition on ARIs, their
impacts on the boundary layer structure, and the discrepancies of these interactions under different
synoptic patterns during pollution processes. The findings hold important implications for effective
management of pollution processes in cloudy and foggy weather.
Key words: Synoptic patterns, Cloud radiation interaction inhibition, Aerosol radiation interaction,
Boundary layer structure, Sichuan Basin.
**1 Introduction**
Particulate matter (PM) pollution has become a significant environmental concern in China (Xie
et al., 2016a; 2016b; Che et al., 2019). High concentrations of aerosols not only worsen air quality
and pose serious health risks to residents, but also have implications for  weather and climate
through their effects on radiation and clouds (Li et al., 2019; Zhao et al., 2020; Alexeeff et al.,
2021; Yang et al., 2021). The interactions between aerosols and clouds present the largest
uncertainty in anthropogenic radiative forcing of the Earth's climate (Liao et al., 2017; Haywood
et al., 2021). Studying interactions among cloud, aerosol and radiation from an air quality perspective
is crucial for a scientific understanding of relationship between weather and pollution.
Excessive emissions are the essential cause of air pollution, with primary aerosol and secondary
aerosol formation playing significant roles in comprehending the complete picture of air pollution
(Peng et al., 2021). Besides, meteorological conditions not only influence on the formation of
secondary aerosols, but also govern the transportation and distribution of both primary and
secondary aerosols, and thereby impact regional and long-range air pollution (Zhu et al., 2018;
Luo et al., 2018; Nichol et al., 2020; Zhang et al., 2020; Jiang et al., 2021). PM and gaseous
pollutants, primarily transported by the planetary boundary layer (PBL), are directly or indirectly
influenced by various meteorological factors such as wind, relative humidity, PBL height (PBLH),
and solar radiation. These factors contribute to the multi–temporal and spatial distribution
characteristics through vertical and horizontal diffusion, physicochemical reactions, and dry and
wet deposition (Park et al., 2017; Shu et al., 2017; Zhan et al., 2019; Huang et al., 2019). Large–
scale synoptic forcing is considered the primary driving condition for meteorological factors, PBL
structure, and the resulting distribution of atmospheric pollutants (Miao et al., 2019; Ning et al.,
2019; Jiang et al., 2020; Li et al., 2021). Specific synoptic patterns can induce advection, which
largely determines the local PBL structure and development. PBL, located at the bottom of the
atmosphere, is responsible for the main exchange of heat, moisture, and matter between the surface
and the free troposphere (Stull, 1988). The fate of pollutants emitted near the surface, a significant
source of aerosols in the air, is largely controlled by the PBL (Garratt, 1994). The PBLH is often
used as a metric to characterize the capacity and dilution of pollutants (Seidel et al., 2010).
Synoptic patterns can directly determine the meteorological conditions of emitted pollutants and
influence their transport by regulating PBL thermal stratification and mechanical turbulence (Stull,
1988; Ning et al., 2018; Zhan et al., 2019; Jiang et al., 2021; Zhang et al., 2022).
Unfavorable meteorological conditions play a significant role in contributing to aerosol pollution.
When pollutants accumulate to a certain degree, aerosols can reduce surface solar radiation
through backscattering or absorbing solar radiation, leading to surface cooling. This decrease in
solar radiation and temperature near the ground weakens turbulent diffusion, suppresses the
convective development of the PBL, and lowers PBLH, which in turn exacerbates aerosol pollution
(Ding et al., 2016; Wang et al., 2018). Moreover, the increase in humidity caused by the decreased
surface saturation vapor pressure and inhibited water vapor diffusion enhances aerosol
hygroscopic growth accelerates liquid–phase and heterogeneous reactions, and contributes to
aerosol pollution (Pilinis et al., 1989). The positive feedback between unfavorable PBL
meteorology and increasing aerosols was found to be responsible for the majority of the increase
in $PM_{2.5}$ during cumulative stages in various regions of eastern China affected by aerosol pollution,
including the North China Plain, the Guanzhong Plain, the Yangtze River Delta, the Two Lakes
Basin, the Pearl River Delta and the Northeast China Plain. But in the Sichuan Bain (SCB), the
feedback is weak due to the suppression of the cloudy mid-upper layer (Zhong et al., 2018; Zhong
et al., 2019). As for the aerosol-cloud interactions, arise from increasing aerosols acting as cloud
condensation nuclei in cloud and translating into larger concentrations of smaller cloud droplets,
leading to an increased cloud albedo reflecting more radiation back to space (Twomey, 1977;
Lohmann and Feichter, 2005). Even a marginal increase in cloud droplets above pristine conditions
in deep convective clouds causes more droplets to reach supercooled levels, which enhances latent
heat release and invigorates convection (Rosenfeld et al., 2009; Possner et al., 2015). Further
increases in cloud droplets result in direct radiative effects, reducing downward solar radiation,
cooling the surface, and inhibiting convection (Scott et al., 2016).
The SCB is surrounded by high mountains with cloudy and wet weather conditions. The mean
annual relative humidity in the SCB is around 75%, with cloud fraction exceeding 80%, and an
average of 1200 hours of sunshine per year. The Chengdu–Chongqing city cluster in the SCB
serves as the economic center of the upper reaches of the Yangtze River in China, accounting for
approximately 10 % of the country's population. However, rapid industrialization and urbanization
in this region have resulted in severe air pollution. The SCB is recognized as one of the most
polluted regions in China, with high black carbon concentrations (Li et al., 2016; Cao et al., 2021).
The Qinghai–Tibet Plateau on the western edge of the SCB significantly influences the transport
and accumulation of pollutants through thermal and dynamic effects (Ning et al., 2017; Shu et al.,
2021). In addition, the Qinghai–Tibet topography leads to higher cloud water content over the SCB
than the other regions (Yu et al., 2004; Yang et al., 2012). Many studies have emphasized the
importance of the interactions between cloud, aerosols and radiation in air pollution processes (Wang et
al., 2018; Hu et al., 2021). High pollutant emissions, combined with the prevalence of cloudy and
foggy weather, make these interactions in the SCB even more complex than those in other regions. The
aerosol radiation interactions (ARI) can be inhibited by cloud in cities like Chengdu (Zhong et al.,
2019). However, there is a lack of in-depth quantitative discussions regarding this aspects in the SCB.
On one hand, the complex terrain in the SCB leads to differences in the meteorological conditions
between them (Ning et al., 2017; Lu et al., 2022). For example, Chengdu is a typical basin city while
Chongqing is a mountain city located on the basin slope, so they have markedly different climate
conditions. It remains to be elucidated whether these conditions will result in spatial disparities in cloud
inhibition on the ARI. On the other hand, synoptic forcing, as the primary driver of meteorological
variations, undoubtedly play an unneglectable role in shaping cloud cover and boundary layer
structures (Miao et al., 2020; Wang et al., 2022; Painemal et al., 2023). The discrepancies in cloud
inhibition on ARI under different synoptic patterns also need to be revealed. Addressing these issues is
crucial for understanding the persistent pollution processes and the intricate interactions between
weather and pollution in the SCB. It holds important implications for the effective management of
pollution processes in cloudy and foggy weather.
Characterized with high aerosol loadings and semi–permanent cloudy weather, the SCB
provides an ideal region for studying the complex interactions between clouds, aerosols, and the
PBL. This study objectively classifies the synoptic patterns influencing the SCB based on long–
term  data. By conducting an integrated analysis of pollutants and meteorological factors, the
primary pollution sources and clean synoptic patterns are identified. To further investigate the
inhibition of cloud radiation interaction (CRI) on ARI under different synoptic patterns in the SCB,
WRF–CHEM simulation experiments are conducted. The results contribute to a deeper
understanding of CRI, ARI, and the PBL interactions in regions influenced by plateau–basin
topography with wet and cloudy weather. The data and methods are presented in Section 2,
whereas Section 3 describes the synoptic patterns and their corresponding impacts on clouds,
aerosols, radiation, and PBL. Finally, the conclusions are presented in Section 5.
**2 Data and method**
**2.1 Observation data**
Air quality monitoring data used in this study were obtained from air quality monitoring sites
established by the Ministry of Ecology and Environment of China across the SCB. Hourly $PM_{2.5}$
observations from 18 stations in the SCB were collected during the winter period from 2015 to 2021
for data analysis and model verification purposes (Fig. 1b). The abbreviations CQ, CD, MY, DY,
LS, MS, YA, ZY, ZG, YB, LZ, NJ, GA, NC, SN, GY, DZ, and BZ represent the following cities:
Chongqing, Chengdu, Mianyang, Deyang, Leshan, Meishan, Yaan, Ziyang, Zigong, Yibin,
Luzhou, Neijiang, Guangan, Nanchong, Suining, Guangyuan, Dazhou, and Bazhong, respectively.

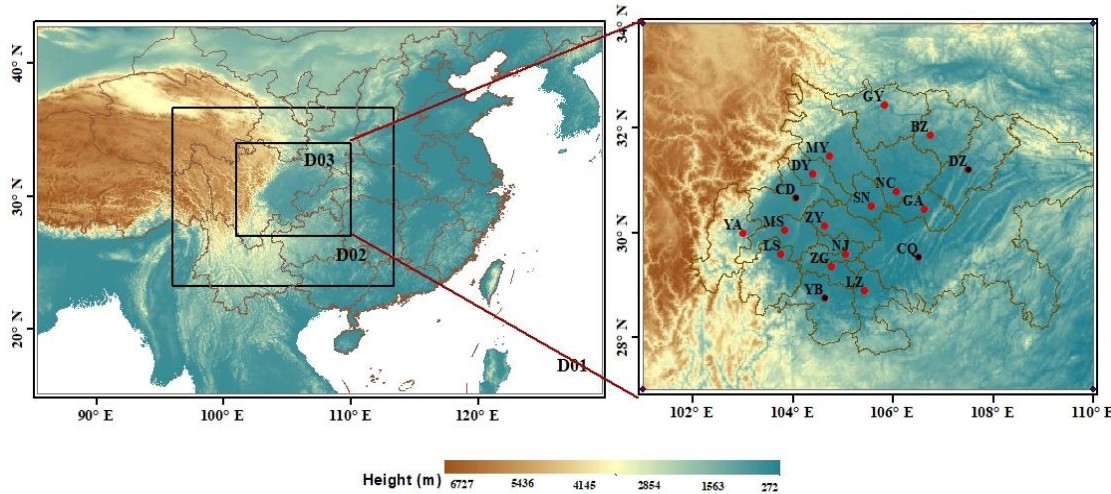

**Figure 1.** (a) Three layers of simulation domains in WRF-Chem with topography map as shading; (b) the locations of 18 air quality monitoring stations (red dots) and 4 sounding stations (black dots) in the Domain 3.

The SCB has four sounding stations: Wenjiang (CD), YB, DZ, and Shapingba (CQ), situated in the western, southern, northwestern, and eastern regions of the basin, respectively (Fig. 1b), and represent different pollution and meteorological conditions in different regions within the SCB. In all, the air pollution over the SCB exhibits a gradual decrease from southwest to northeast. Statistical analysis indicates that the western and the southern basin experience the most severe pollution. The western basin shows the highest pollution proportion, while the southern basin exhibits the highest occurrence of heavy pollution. In the northeastern basin, specifically in DZ, heavy pollution is more likely to occur during winter, which verifies it to be the third highest pollution zone outside the western and southern basin. This makes the spatial distribution during winter differs from the overall annual pollution pattern in the SCB (Lu et al., 2022; Qi et al., 2022). Regarding meteorological conditions, research reveals that DZ has the lowest ventilation coefficient during winter, while CQ has the highest. The SCB experiences frequent temperature inversions, with CD having a higher occurrence of inversions compared to the other three cities. CD also exhibits the strongest inversion intensity and is prone to multi-layer inversions. On the other hand, YB and CQ have greater inversion thickness, while CD has the smallest inversion thickness (Feng et al., 2020). The vertical distribution of the meteorological factors used in the study was obtained from an L–band sounding radar, collecting temperature, pressure, humidity, and wind data at 00:00 and 12:00 Coordinated Universal Time (UTC) on vertical levels every second from the surface up to 30 km. Ground observation data from the four cities, including temperature and dew point temperature, were used for meteorological factor simulation verification. All meteorological data were obtained from the China Weather Website Platform maintained by the China Meteorological Bureau. As for the calculation of PBLH, there are various methods to determine the PBLH, and differences in methods, data or threshold values may yield quite different PBLH results (Seibert et al., 2000; Eresmaa et al., 2006; Jiang et al., 2021). The bulk

Richardson number (*Ri*) method was adopted to calculate the PBLH with sounding data in
the study by assuming that the PBLH is the height at which *Ri* reaches its critical value (*Rc*).
*Ri* at a certain height *h* is calculated as follows:
$$Ri = \frac{(g/\theta_{v0})(\theta_{vh} - \theta_{v0})h}{u_h^2 + v_h^2}$$

Where g is the acceleration of gravity, $\theta_{v0}$ and $\theta_{vh}$ are the virtual potential temperature at surface
and the height *h*, respectively, and $u_h$ and $v_h$ are the meridional and zonal wind components at *h*.
We adopted the *Ri* method and *Rc* to be 0.25, because the EAR5 and YSU schemes use the same
method  and threshold value when calculating PBLH (Hong et al., 2006; ECMWF, 2017).
CD, YB, DZ and CQ were selected as representative cities for analysis in the study. These four
representative cities are located in the western, southern, northwestern, and eastern regions of the basin,
to capture diverse pollution and meteorological conditions within the SCB. These cities were chosen to
represent the most polluted regions (Zhao et al., 2018; Lu et al., 2022), as well as typical basin and
mountainous cities. Furthermore, there are only four sounding stations in the SCB available, which are
located in these four representative cities. They can provide valuable vertical and surface
meteorological observations, as well as pollution data, contributing comprehensive dataset used in this
study.
ERA5 reanalysis data from the European Centre for Medium-Range Weather Forecasts (ECMWF),
which assimilates comprehensive observation data, including ground observation, sounding data,
aircraft observation data, and satellite observation data, were obtained for synoptic pattern
classification and their impact on meteorological factors in four representative cities. The EAR5
data at the 850 hPa pressure level were collected for the synoptic pattern study. Additionally, cloud
liquid water content and downward solar radiation derived from the EAR5 single–level datasets
were obtained to assess the influences of synoptic forcing on CRI studies, while PBLH were
adopted to conduct the simulation verification. Previous studies have demonstrated the reliability
of ERA5 data in estimating cloud properties, including the cloud liquid content (Yao et al., 2019;
Nandan et al., 2022; Ojo et al., 2023).

## 2.2 Synoptic pattern classification

The objective classification was conducted on the synoptic patterns of the SCB using ERA5 data,
including geopotential height, u, and v components of winds at the 850 hPa pressure level. The
analysis covered an area of 97–117° E and 24–37° N with a horizontal resolution of 0.25° × 0.25°.
Given that PM pollution in the SCB is primarily prevalent during winter months (Zhao et al., 2018;
Lu et al., 2022), the synoptic pattern classification was performed for winter seasons from 2015 to
2021 (December, January, and February) using the principal component analysis in the T–model
(T–PCA) objective method. Compared with the subjective classification method, the objective
method can process large amounts of data without relying on subjective experience (Huth et al.,
2008; Miao et al., 2017). Among various classification methods, the T–PCA method accurately
reflects the characteristics of the original synoptic circulations and exhibits spatial and temporal
stability (Huth et al., 1996; Huth et al., 2008). Consequently, the T–PCA has been widely used in
synoptic pattern classification researches (Ning et al., 2019; Miao et al., 2020; Li et al., 2021).

## 211 2.3 Model configuration and simulation experiments

To understand the combined effects of synoptic patterns and CRI inhibition on ARI and PBL, a
series of parallel experiments were conducted on the simulation of a typical pollution episode
using the Weather Research and Forecasting model with Chemistry (WRF–Chem v3.9.1) (Grell et
al., 2005). The Advanced Research WRF (ARW) dynamics solver integrates the compressible,
nonhydrostatic Euler equations, for example, the momentum equation, the continuity equation, the
thermodynamic equation, the moisture equation and the ideal-gas equation of state (Skamarock et
al., 2008). The model domain (Fig. 1a) was centered over the SCB and utilized three layers of
nested grids with horizontal resolutions of 27, 9, and 3 km, respectively. A total of 32 vertical
layers spanning from the surface to 100 hPa were defined. Initial and boundary meteorological fields
were obtained from the National Centers for Environmental Prediction Final reanalysis data with a
horizontal resolution of $1° \times 1°$ and 6 h time interval. For chemical process simulations,
anthropogenic emissions were sourced from the Multiresolution Emission Inventory for China
(MEIC) in 2016, featuring a grid resolution of $0.25° \times 0.25°$. To address the empirically
overestimated $PM_{2.5}$ emissions by the MEIC in the SCB (Zhan et al., 2023), the ensemble square
root Kalman filter were implemented on the $PM_{2.5}$ emission during simulation (Wu et al., 2018;
Lu et al., 2021). Biogenic emissions were calculated online using the Guenther scheme (Guenther
et al., 2006). Table 1 provides a summary of the chosen physical and chemical parameterization
schemes. The parameterization schemes employed in this study is the one used by the Chongqing
Meteorological Bureau in the daily operational activities. The schemes have been obtained
through multiple sets of control experiments and are considered suitable for the simulation in the
SCB.
**Table 1 The main options of WRF–Chem**

| Items | Contents |
| --- | --- |
| **Domains (x, y)** | (155, 110), (184, 160), (320, 250) |
| **Grid spacing (km)** | 27, 9, 3 |
| **Center** | (29.1° N, 106.2° E) |
| **Time step (s)** | 60 |
| **Microphysics** | WRF Single–Moment 5 class (WSM5) scheme |
| **Longwave radiation** | RRTMG scheme (Iacono et al., 2008) |
| **Shortwave radiation** | RRTMG scheme (Iacono et al., 2008) |
| **Planetary boundary layer** | Younsei University scheme (Hong et al., 2006) |
| **Land surface** | United Noah land surface model (Tewari et al., 2004) |
| **Cumulus parameterization** | Grell–Freitas ensemble scheme |

| | | |
|---|---|---|
| | (Grell et al., 2013) | |
| **Advection** | fifth- and third-order differencing for horizontal and vertical advection respectively | |
| **Photolysis scheme** | Fast-J photolysis (Fast et al., 2006) | |
| **Gas–phase chemistry** | RADM2 (Stockwell et al., 1990) | |
| **Aerosol module** | MADE/SORGAM (Schell et al., 2001) | |


To assess the impact of CRI inhibition on ARI under typical synoptic pollution patterns, four
parallel experiments were conducted using simulation models. The selected simulation period for
these experiments was January 1-7, 2017. The period was selected for two reasons: the Chinese
government announced clean-air action in the year of 2013, aiming to reduce $PM_{2.5}$ concentrations in
the next 5 year. Specifically, the year of 2017 was identified as a key year for assessing $PM_{2.5}$ pollution
in China, as significant  practical actions were implemented during the period (Wang et al., 2020) and
the selected period encompassed both typical pollution and clean weather patterns.
The baseline experiment (BASE) included both CRI and ARI in the simulations. In contrast, the
three sensitivity experiments focused on excluding either ARI or CRI. Experiment 1 (EXP1) did not
consider ARI, Experiment 2 (EXP2) did not include CRI, and Experiment 3 (EXP3) omitted ARI
when CRI was not included. The differences between BASE and EXP1 represented the
disturbances caused by ARI, while EXP2 and EXP3 represented the influences of ARI without
CRI inhibition. Detail differences between the experiments could be found in Table 2. The
numerical experiments were initiated at 00:00 UTC on December 30, 2016, and ran until 00:00
UTC on January 8, 2017, with the first 48 hours designated as a model spin–up period.
**Table 2 Four numerical simulation experiments are conducted in the study**

| Experiments | Description | Results | Meaning |
|---|---|---|---|
| **BASE** | Baseline simulation | BASE-EXP1 | Disturbances by ARI |
| **EXP1** | Only shutting down ARI | | |
| **EXP2** | Only shutting down CRI | EXP2-EXP3 | Influences of ARI without CRI |
| **EXP3** | Shutting down both ARI and CRI | | |

*ARI: aerosol radiation interaction; CRI: cloud radiation interaction

**3 Results and discussions**
**3.1 Relationships between synoptic patterns and $PM_{2.5}$ pollution in the SCB**
Figure 2 illustrates the daily mean variations in $PM_{2.5}$ concentration and vertical distributions
of potential temperature (PT) during winter period from 2015 to 2021, with a focus on the pollution
episodes. The four sounding stations located in separate areas of the SCB (CD, YB, CQ, and DZ),
consistently experienced pollution processes characterized by simultaneous changes in vertical
thermal structures. For example, during the pollution events in January 2017 and December 2020,
the PM$_{2.5}$ concentrations in all four cities reached their peak levels at the same time before rapid
declining (Fig. 2). Interestingly, these pollution episodes were accompanied by warming in the
upper layer atmosphere, while a decrease in PM$_{2.5}$ concentration correlated with cooling. Despite
the significant distances between these cities (approximately 200–400 km), the synchronized
changes in pollutant concentrations and vertical thermal structures could be attributed to large–
scale synoptic patterns (Miao et al., 2020; Li et al., 2021). While the four cities with sounding
stations were selected as representatives for vertical thermal structure analysis, other cities in the
SCB also experienced pollution episodes and relevant physical processes, except for GY (Fig. S1).
GY is located in the northern edge of the SCB, bordering Shaanxi and Gansu Provinces. The proportion
of heavy PM$_{2.5}$ pollution in GY is the lowest in the basin, but the proportion of PM$_{10}$ pollution is higher
than other cities of SCB (Lu et al., 2022). Due to the lower PM$_{2.5}$ concentration, the two pollution
processes in January 2017 in GY were not as significant as in other cities whithin the basin. However,
the warming of upper air coincided with PM$_{2.5}$ increase could still be observed.

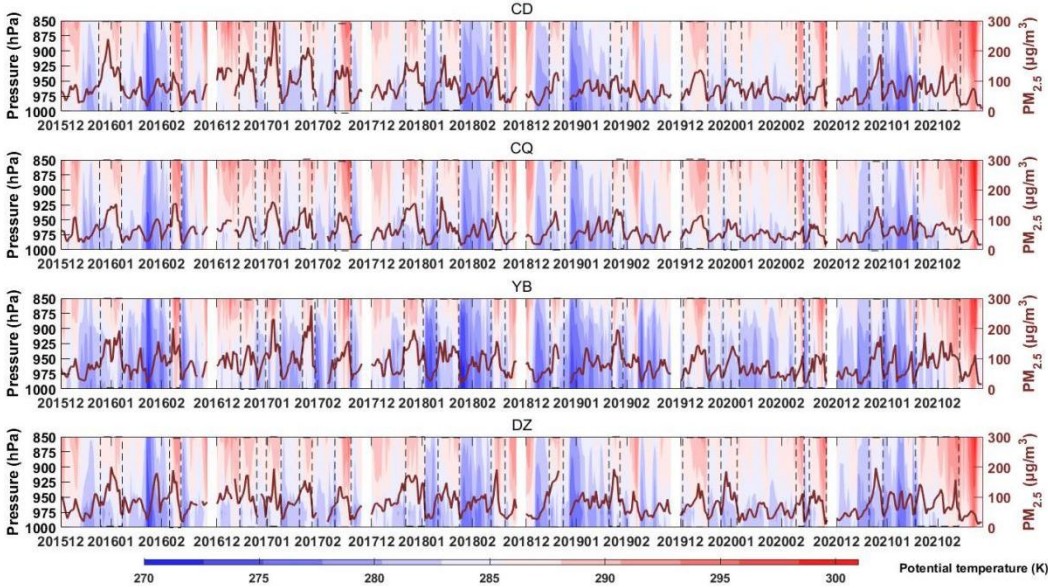

**Figure 2.** Time series of daily mean PM$_{2.5}$ and potential temperature derived from the sounding data during 2015-2021 winter months. The PM$_{2.5}$ pollution episodes are marked with black dotted boxes.


The time series of daily mean PM$_{2.5}$ from air quality monitoring sites and the accompanying
vertical distributions of temperature, relative humidity, and wind in CD, CQ, YB, and DZ derived from
the sounding stations are shown in Fig. S2, focusing on January 2017 as an example for analysis.
During this month, two severe PM$_{2.5}$ pollution episodes occurred: one from January 1 to 7 and another
from January 24 to 31 in 2017. These pollution episodes had a significant impact on air quality in all
four cities. The highest daily PM$_{2.5}$ concentrations recorded during these episodes were 291.17 μg/m$^3$ in
CD and 276.2 μg/m$^3$ in YB. Pollution in early January exhibited a gradual increase in PM$_{2.5}$ levels from
January 1 to 3, with upper air warming and the emergence of an inversion above the PBL. Additionally,
lower humidity and higher wind speeds above 1500 m were observed during the pollution
accumulation period. Similarly, the late January pollution episode showed a rapid increase in PM$_{2.5}$,
from January 24 to 27, together with warming, dryness, and high wind speed above 1500 m in all four

cities. These consistent meteorological conditions during the pollution periods indicated significant synoptic forcing. The previous study has found that winter heavy pollution processes in the SCB are usually associated with abnormal warming above the 850 hPa (Lu et al., 2022). The warming is induced by strong southerly airflow above the basin. The southerly airflow in winter over the SCB originates from the Yunnan-Guizhou Plateau or the Indian Peninsula, characterized with high temperature, dryness, and high wind speed. The strong southerly airflow forms a warm lid over the basin, suppressing the vertical exchange of pollutants within the basin. As a result, pollutants accumulate rapidly, which may explain the phenomenon of rapid $PM_{2.5}$ growth accompanied by warming, dryness, and strong winds above 1500 m. Notably, the key layer for studying the connection between synoptic patterns and $PM_{2.5}$ pollution is approximately 850 hPa, corresponding to a height of approximately 1500 m within the PBL, where changes in specific meteorological conditions primarily affect surface–emitted pollutants.

Using ERA5 reanalysis data for winter (December, January, and February) from 2015 to 2021, the 850 hPa synoptic patterns over the SCB were objectively classified into six types (Fig. 3). According to the relative positions of the high–pressure and low–pressure systems in the basin, these synoptic patterns could be described as follows: (1) strong high pressure in the north, (2) east high west low (EHWL) pressure, (3) weak high pressure in the north, (4) weak ridge of high pressure after the trough, (5) low trough (LT), and (6) strong high pressure. Patterns 1 and 3 exhibited high pressure in the northern SCB, which differed from the high–pressure intensity. With strong high pressure, the basin was primarily controlled by northerly airflow. Under weak high–pressure conditions, the basin was dominated by an easterly backflow. Patterns 2 and 5 had high and low pressures near the basin, forming a relatively dense isopotential altitude gradient and resulting in strong southerly winds over 850 hPa. Pattern 4 was a weak high–pressure ridge after a trough controlled the SCB with sparse isobaric lines and weak winds leading to static and stable weather conditions. During Pattern 6, the SCB was controlled by the cold high-pressure system, accompanying weak northerly airflow on the basin. Pattern 6 usually evolved from either Pattern 1 or Pattern 3.

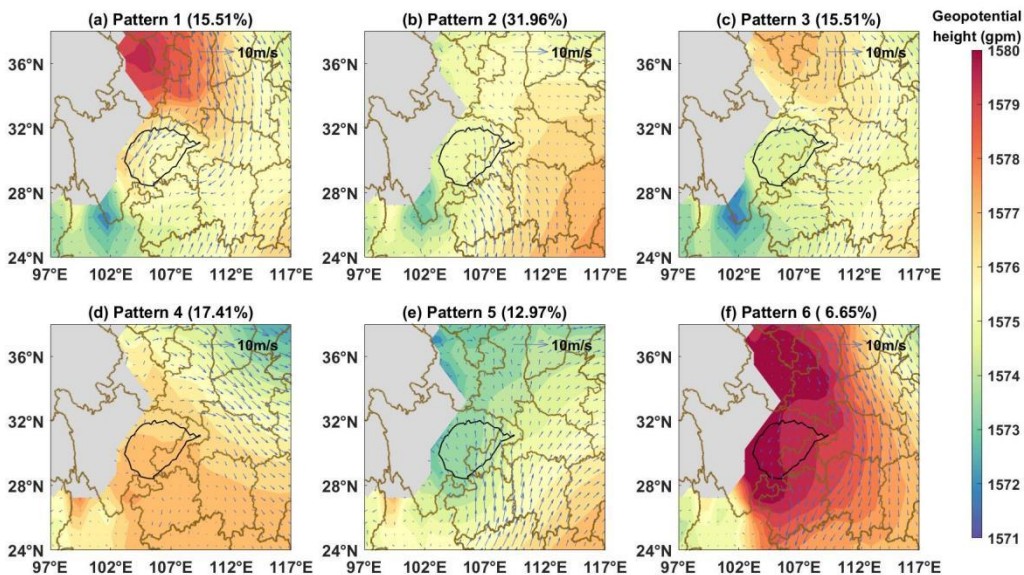

**Figure 3.** The 850hPa geopotential height field (shading) with wind vector fields (blue vectors), and frequency of occurrence for 6 synoptic patterns during 2015-2021 winter months. The SCB was outlined with an altitude contour of 750 m terrain height (black lines).


Patterns 2, 4, and 5 exhibited higher frequencies of pollution occurrence (PM$_{2.5}$ daily
concentration $\geq$ 75 μg/m$^3$) according to statistical results from 18 cities in the SCB during the
2015–2021 winters (Fig. 4a). These patterns were associated with high PM$_{2.5}$ concentrations in 50–
70 % days, including CD, DY, and MY in the northern SCB, 40–60 % for cities in the southern
SCB, such as ZG and YB, and also 40–60 % of days for cities in the northern SCB, such as CQ, DZ,
NC, and GA. Furthermore, the average PM$_{2.5}$ concentrations in the respective cities for the six
synaptic patterns were calculated (Fig. 4b), aligning with the frequency of pollution occurrence.
The days under Patterns 2, 4, and 5 exhibited higher average daily PM$_{2.5}$ concentrations. The average
concentrations under these three synoptic patterns were 99.19, 103.43, and 111.97 μg/m$^3$ for CD,
95.44, 87.98, and 94.26 μg/m$^3$ for YB, 79.14, 83.96, and 74.77 μg/m$^3$ for CQ, and 91.02, 104.64,
and 91.51 μg/m$^3$ for DZ, respectively. Regarding the impact of synoptic patterns on the
accumulation or dispersion of PM$_{2.5}$, Fig. 4c illustrates the average daily changes in PM$_{2.5}$
concentration compared with the previous day for CD, CQ, YB, and DZ under the six synoptic
patterns. Patterns 2 and 5 exhibited the most significant PM$_{2.5}$ accumulation under the influence of
southerly airflow. The average PM$_{2.5}$ concentration under Pattern 1, 3 and 6 was lower in all cities
of SCB than other three pollution patterns (Fig. 4a). Besides, the day to day PM$_{2.5}$ variations under
Pattern 1, 3 and 6 exhibited negative growth trend in the four representative cities (Fig. 4c). As a
result, Pattern 1,3 and 6 were identified as the "clean pattern". In addition, the pollution
occurrence frequency of which was found higher for cities located in the eastern part of the SCB
than other parts. Under Pattern 6, strongest northerly airflow affects the basin. The eastern part of
the basin consists of parallel ridges and valleys, which reduces wind speed. The stronger the wind
is, the more obvious the reduction of wind by terrain is. In contrast, the western part is relatively
flat, which can result in higher surface wind speeds. The difference in wind impacted by terrain
led to a weaker pollution removal effect in the eastern region, thus contributing to a higher
proportion of pollution days under Pattern 6. Besides, differences in precipitation rates between
eastern cities and other regions were not significant (the proportion of rainfall with a daily
accumulated precipitation exceeding 10 mm in CD, CQ, YB and DZ under Pattern 6 were all less than
3%) , which might not the main reason why eastern cities in the SCB experience higher pollution
frequency.

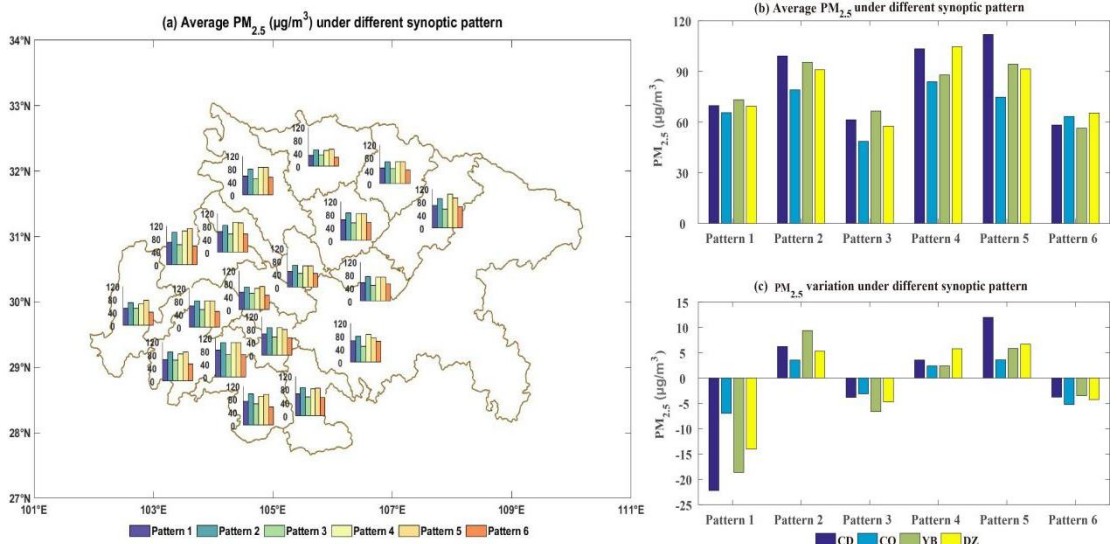

**Figure 4.** (a) The pollution occurrence frequency at 18 air pollution stations in SCB, (b) (c) average PM$_{2.5}$ concentrations and PM$_{2.5}$ day to day variations at 4 representative SCB cities, under 6 synoptic patterns.


The time series of daily mean PM$_{2.5}$ and the day-to-day classification of 850 hPa synoptic
patterns are shown in Fig. S3, from December 2016 to January 2017. Six pollution episodes
occurred during this period (December 03–12 and 16–26, 2016; January 1–7, 16–19, and 20–28,
2017; and February 14–23). It is observed that pollution episodes consistently began with Pattern 2
and ended with Pattern 1, accompanied by a rapid decline in PM$_{2.5}$. This finding suggests that
Pattern 2 acted as a key synoptic forcing for the initation of pollution episodes. Additionally,
statistical results revealed that Pattern 2 accounted for a high proportion of PM$_{2.5}$ increase during the
six pollution episodes, reaching 48.48 %, while Pattern 5 had the second highest proportion of
21.88 %, with Patterns 2 and 5 combined accounting for more than 70 % of the pollution episodes.
For example, during the two heavy pollution events in early and late January 2017, PM$_{2.5}$ rapidly
accumulated with the interplay of Patterns 2 and 5. These two patterns represented a substantial
proportion of 31.96 % and 12.97 %, respectively, during winters from 2015 to 2020 at 850 hPa
level in the SCB (Fig. 3). Based on this analysis, Patterns 2, 4, and 5 were identified as synoptic
pollution patterns, whereas Patterns 1, 3, and 6 were as clean patterns. In summary, Patterns 2 and
5 played crucial roles in the initiation and accumulation of PM$_{2.5}$ during pollution episodes.
The discussion above showed that pollution in the SCB tended to occur when southerly airflow
controlled the upper-layer of the basin (Pattern 2 and 5), while the dispersion of pollutants was
accompanied by northerly winds, which aligns with the findings of Lu et al. (2022). This study
indicated that southerly airflow in the upper-layer could bring warm air, leading to warming above
the basin and forming a "warm lid". The surrounding mountains and plateau with the "warm lid"
contributed to the formation of a relative enclosed space within the SCB, facilitating local
circulations and allowing for the thorough mixing and secondary reactions of local emission and
pollutants transported from outside. As a result, persistent and severe pollution often occurred
under the influence of southerly airflow. When the northerly airflow began to dominate the SCB,
the "warm lid" and local circulation were disrupted, leading to dispersion of pollutants through
advection and vertical transport. Northerly winds were often associated with cold air and
sometimes accompanied by weak precipitation, resulting to wet deposition and the removal of
pollutants. Therefore, the arrival of northerly airflow often signified the ending of the pollution
episode. The evolution of 850 hPa synoptic forcing and vertical meteorological conditions (Fig. 2
and 6) aligns with the study of Lu et al (2022). Therefore, there are also similar pollution change
mechanisms.

Due to the convergence of air moving eastward across the Tibetan Plateau, the SCB experiences

frequent wet and cloudy weather, with cloud cover fraction exceeding 80 % (Yu et al., 2004;
Zhang and Lin, 1985). Clouds undoubtedly play an unneglectable role in the interactions of
aerosols, radiation, and the PBL under typical synoptic forcing in this region. This study evaluated
the average cloud liquid water content, downward solar radiation, and PBL under the influence of
the six classified synoptic patterns in CD, CQ, DZ, and YB, using data from ERA5 (Fig. 5). The
reanalysis data revealed significant higher cloud liquid water contents with Patterns 2 and 5, likely
triggered by robust southerly air prevailing at 850 hPa over the SCB (Fig. 3). This southerly air
brought warm and moist air, contributing to cloud formation. Dense clouds reduced solar radiation
through reflection and scattering, resulting in surface cooling and inhibiting PBL development.
The PBLH under Patterns 2 and 5 was approximately 900–1000 m, lower than that under the
influence of clean synoptic Pattern 6 at 1500 m or Pattern 1 and 3 at 1200–1300 m (Fig. 5). In
contrast, the clean synoptic Pattern 1 was characterized by a strong northerly flow at 850 hPa,
resulting in lower cloud liquid water content over the basin and increased solar radiation, promoting
PBL development. The lower PBLH with more stable stratification caused by the CRI in Patterns 2
and 5 could partially explain the rapid accumulation of $PM_{2.5}$ during these two pollution patterns.

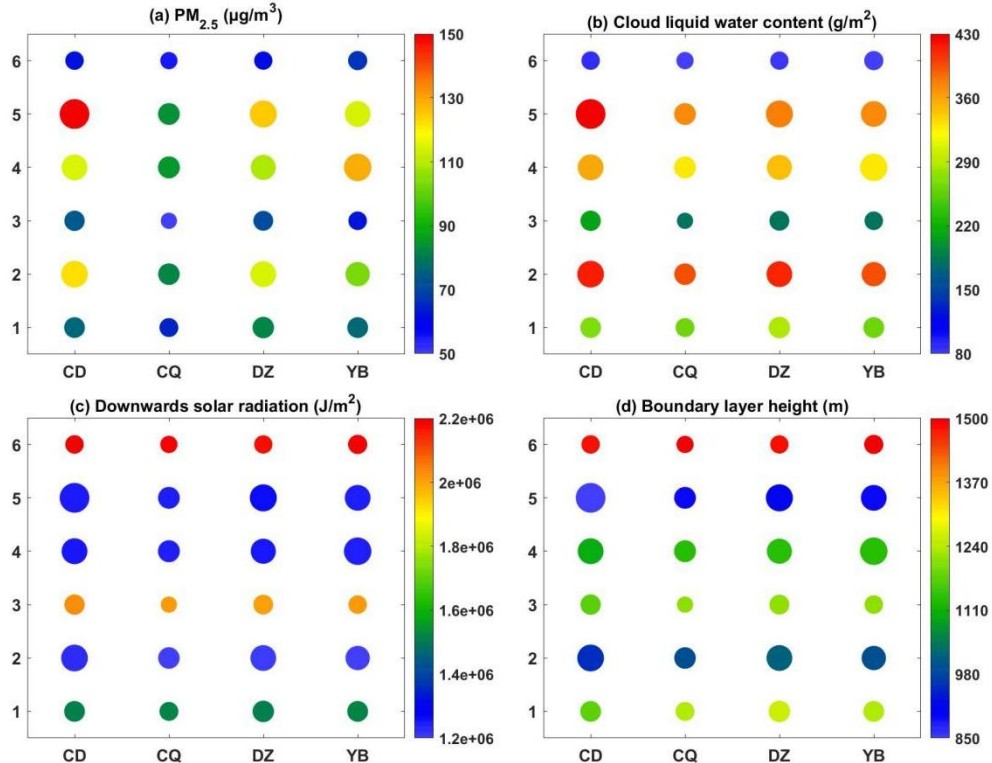

**Figure 5.** The averaged (a) PM₂.₅ concentrations, (b) cloud liquid water contents, (c) downwards solar radiation and (d) boundary layer height derived from 2015-2021 winter months ERA5 reanalysis data, at 4 representative SCB cities under 6 synoptic patterns, the dot sizes represent PM₂.₅ concentrations.


## 3.2 Integrate impacts of synoptic patterns and the CRI inhibition on ARI

Based on the above analysis, Patterns 2 and 5 were identified as the key pollution synoptic
patterns accompanying dense clouds and, thus, strong CRI. However, the effects of pollution
patterns on ARI and their interaction with CRI in the SCB remain unclear and warrant further
investigation. A typical pollution episode from January 1–7, 2017, was selected to understand these
complex processes and simulated using WRF–Chem. The BASE simulations were verified with
observations to determine the accuracy and reliability of the simulation results. The simulation and
observation of PM₂.₅, T2, TD2 and wind speed values with some statistical metrics in CD from
January 1–7 are shown in Fig. 6a-d. Similar information at CQ, YB and DZ can be found in the
Fig. S4. The MB of the simulated and observed PM₂.₅ concentrations were -15.59, -13.42, 2.10
and -13.11 μ g/m³, with NMB values of -4.12%, -4.22%, 6.01% and -0.68% at four cities,
respectively, which are within the acceptable standards (NMB < ± 15%). The R of PM₂.₅ were
78.91%, 57.23%, 61.15% and 62.86% for four representative cities, respectively. The statistical
metrics for PM₂.₅ are consistent with previous studies (Wang et al., 2020; Shu et al., 2021; Zhan et
al., 2023), indicating that our model results for PM₂.₅ are reasonable and acceptable. Regarding to
the surface meteorological factors, low MB and high R for both temperature and dew point
temperature suggested good simulation performance for these variables. However, the simulation
results for wind speed were poor, which was expected under conditions of low wind and complex
terrain. The high observed calm wind frequency, influenced by the starting speed of the
anemometer, led to an overestimation in the simulation (Shu et al., 2021; Zhan et al., 2023).
Additionally, it could be argued that unresolved topographic features introduce additional drag,
beyond that generated by vegetation, which was not considered in the WRF model (Jimenez and
Dudhia, 2012).
In addition, the temporal averaged and variations of vertical profiles for potential temperature,
relative humidity  and wind speed in the model were compared with the sounding data in CD (Fig.
6e-m). Model evaluation of vertical structures in CQ, YB and DZ can be found in Fig. S5. The
SCB is characterized with cloudy and foggy conditions, which result in abundant water vapor and near
100% relative humidity above the nocturnal boundary layer. Models often underestimate the humidity
above the boundary layer during night in the SCB (Shu et al., 2021). Furthermore, due to complex
terrain and measurement bias of the anemometer for weak winds, the evaluation of  simulation results
for wind speed often exhibit certain deviations (Jimenez and Dudhia, 2021; Shu et al., 2021; Zhan et al.,
2023). For the verification of PBLH, sounding data are commonly regarded as reliable vertical
observation records, and PBLH calculated based on sounding data can be used as the true values to
compare with other data for long-term validation (Guo et al., 2016). However, for short-term studies,
due to limited availability of sounding data at only 00:00 and 12:00 UTC, the ERA5 data were also
incorporated for the model evaluation of PBLH in this study (Fig. 6 and Fig.S5). The simulation PBLH
showed a consistent trend with those calculated from ERA5 and sounding data. Overall, the simulation
results can capture the meteorological and $PM_{2.5}$ variation trends. According to the simulation
evaluation standards for the SCB in previous studies (Wang et al., 2020; Zhan et al., 2023), the results
is acceptable and reasonable; thus, the simulation can be used for subsequent analysis and discussion.

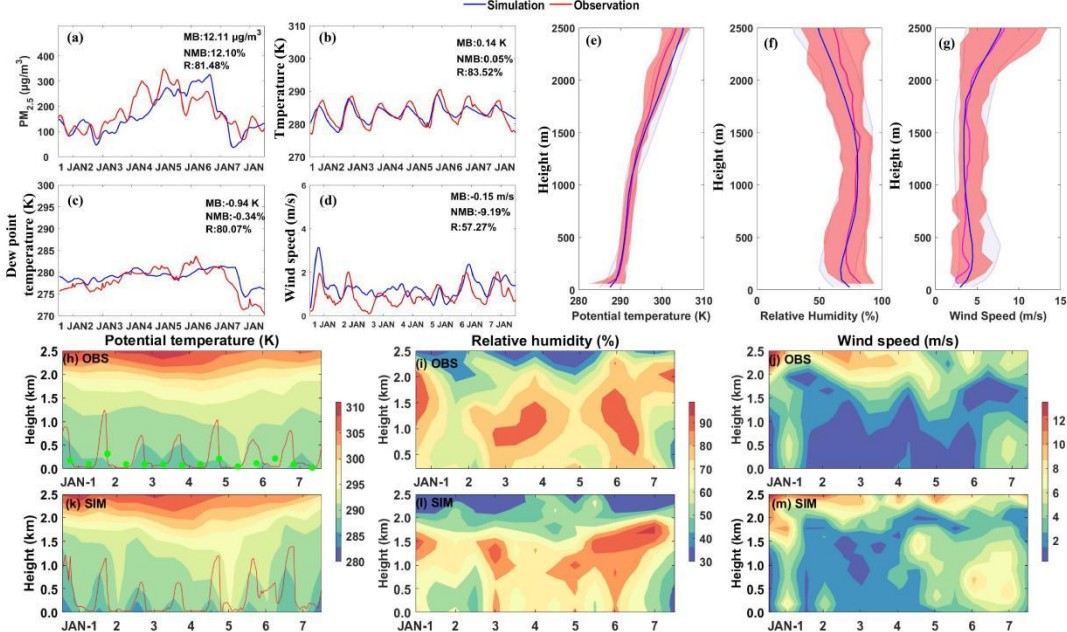

**Figure 6.** Time series of hourly simulated and observed (a) PM$_{2.5}$ concentration, (b) temperature at 2 m, (c) dew temperature at 2m and (d) wind speed near surface, and comparison of simulated and observed mean vertical profile of (e) potential temperature, (f) relative humidiy and (g) wind speed, the red and grey shaded areas represent deviations to the mean values of observation and simulation, respectively. The simulated and observed time-height sections of (h)(k) potential temperature, (i)(l) relative humidity and (j)(m) wind speed are also given, while the red lines in (h)(k) are time series the boundary layer heights derived from ERA5 data and simulation with green dots representing boundary layer heights calculated with sounding data. The above figures display information in CD. Additionally, the model verification information regarding CQ, YB and DZ can be found in Supplement Figure 3 and 4.


During the pollution episode that occurred from January 1 to 7, 2017, the pollution synoptic
patterns controlled the SCB as follows: Pattern 2 from January 1 to 3, Pattern 5 from January 4 to 6,
and Pattern 1 on January 7. Consequently, PM$_{2.5}$ pollution in the SCB occurred on January 1–6 and
rapidly dissipated on January 7 (Fig. 7). The mean geopotential height at 850 hPa derived from the
simulation of January 1–3 under Pattern 2 showed EHWL, with southerly flow prevailing over the
SCB (Fig. 7a). The resulting upper air warming suppressed PBL development (Fig. 7d). During
January 1–3 under Pattern 2, the average PBL heights were lower (Fig. 7d), acting as a lid above the
SCB and hindering the airflow within the basin due to the surrounding mountains. Low wind
speeds provided adverse diffusion conditions for pollutants emitted into the basin, resulting in
severe pollution in the western and southern SCB (Fig. 7g). From January 4 to 6, the low pressure
over the SCB evolved into a LT pattern, termed Pattern 5 in the previous analysis. Compared with
Pattern 2, the isobaric lines were denser under the influence of the LT, leading to stronger southerly
winds above the SCB (Fig. 7a–b). Lower average PBL heights were observed during January 4–6

under Pattern 5 compared with those of January 1–3 under Pattern 2 (Fig 7d–e), primarily due to stronger upper air warming and more stable stratification. Pollutants that accumulated during January 4–6 from the earlier pollution episode (January 1–3) further increased (Fig. 7g–i). On January 7, high pressure in the north dominated the SCB, with a prevailing northerly flow over the basin (Fig. 7c). The PBL height quickly increased due to upper-layer cold advection (Fig. 7f), resulting in a rapid decrease in PM$_{2.5}$ (Fig. 7i). Overall, synoptic patterns play a key role in the accumulation and diffusion of PM$_{2.5}$ during pollution episodes by modulating PBL development and stratification stability.

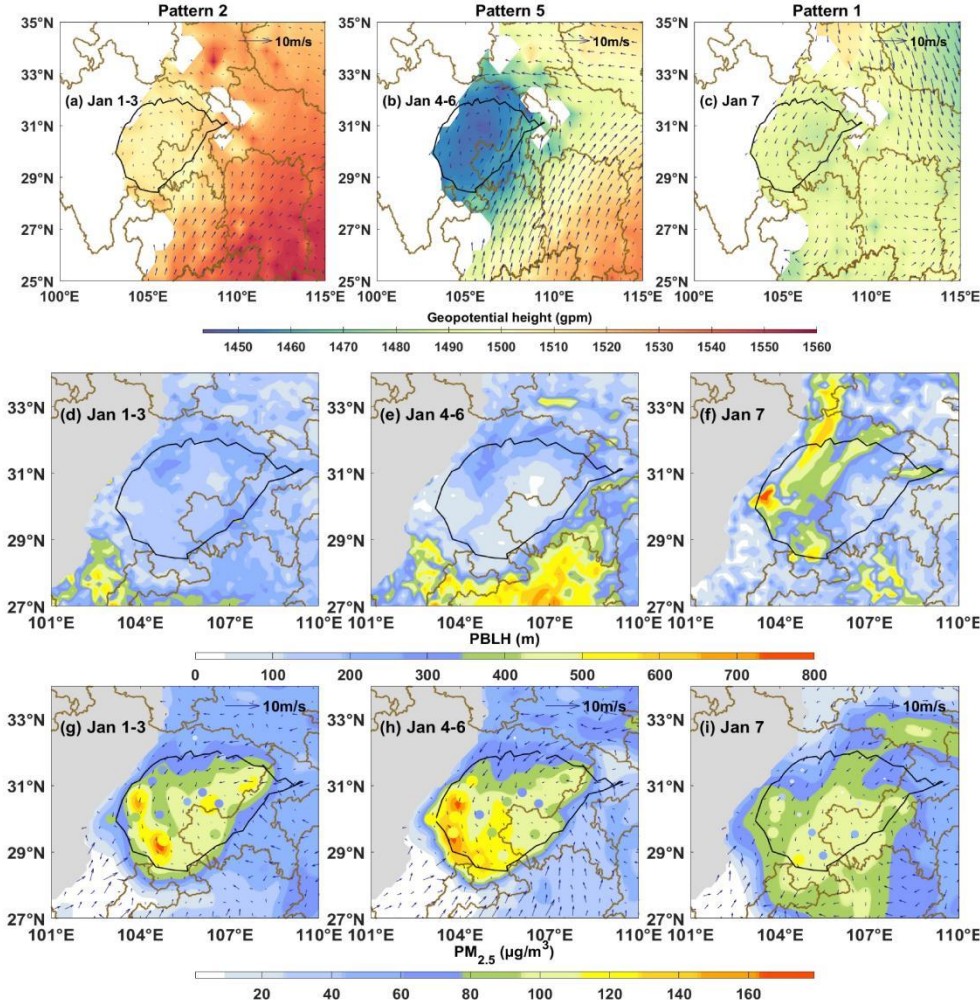

**Figure 7.** The simulated (a)-(c) 850 hPa geopotential height field (shading) with wind vector fields (blue vectors), (d)-(f) boundary layer height and (g)-(i) PM$_{2.5}$ concentrations (shading) and wind vector fields at 900hPa (blue vectors) for 1-3, 4-6 and 7 January. The size and color of scatters in (g)-(i) show corresponding observed PM$_{2.5}$ concentrations at 18 air quality monitoring stations. The SCB was outlined with an altitude contour of 750 m terrain height (black lines).

Pollutant accumulation can regulate the PBL structure through the ARI, further exacerbating pollution (Wang et al., 2018; Miao et al., 2020). In the SCB, this positive feedback is weaker than in the other regions and may be inhibited by cloud radiation (Zhong et al., 2019). A series of simulation experiments were conducted to investigate the aerosol radiation feedback in the SCB

under the influence of two typical synoptic pollution patterns, as described in Section 2.4. BASE–
EXP1 represents the perturbations caused by ARI, whereas EXP2–EXP3 demonstrates changes
through ARI without CRI inhibition. Aerosols led to surface cooling through absorbing and
scattering solar radiation, thereby inhibiting the development of the PBL, which in turn facilitated
pollutant accumulation (Fig. 8). Compared with Pattern 2, the aerosol concentrations in Pattern 5
were higher, resulting in greater reduction of downward solar radiation reduction due to ARI,
leading to more pronounced cooling near the ground and a lower PBLH. Overall, the ARI in Pattern
5 was more significant than that in Pattern 2, regardless of CRI inhibition (Fig. 8).

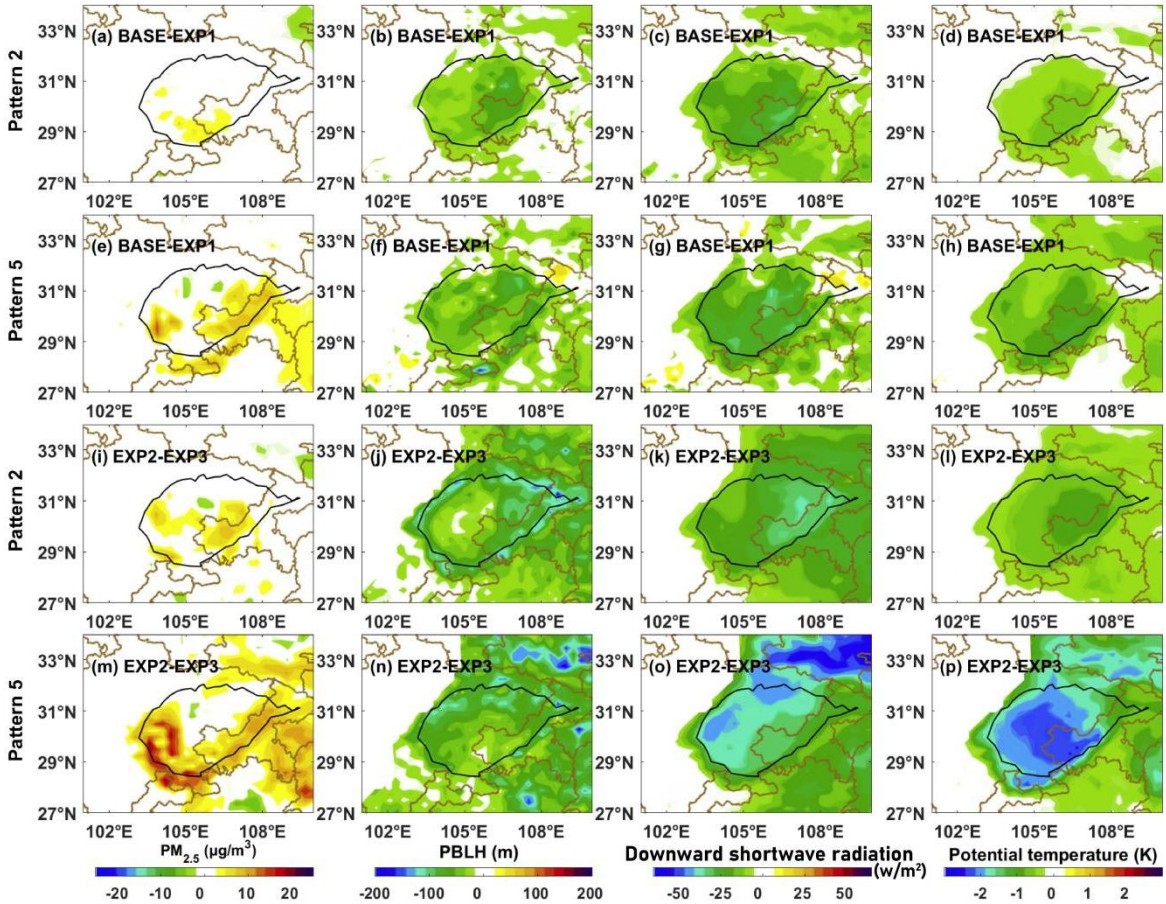

**Figure 8.** Spatial distribution of perturbations induced by (a)-(h) aerosol radiation interactions
(ARI), and (i)-(p) ARI without the cloud radiation interaction (CRI) inhibition during 1-3 and 4-6
January representing Pattern 2 and Pattern 5 synoptic forcing, respectively. The SCB was outlined
with an altitude contour of 750 m terrain height (black lines).


Furthermore, parallel simulation experiments revealed that the CRI significantly attenuated the
ARI in the SCB under both pollution synoptic patterns. When the CRI was not considered, more
solar radiation penetrated the PBL. Dense aerosols accumulated near the surface, intercepting more
downward shortwave radiation, and resulting in stronger cooling near the ground. This suppressed
the development of the PBL, and contributed to a more remarkable ARI (Fig. 8). Regarding the
horizontal spatial distribution, a strong ARI was primarily observed in CQ, as well as in the western
and southern SCB, despite CQ experiencing lower pollutant concentrations compared to the other
two regions (Figs. S4 and 7). This weaker ARI phenomenon in the western SCB was also reported
by Zhong et al. (2019) and was attributed to CRI inhibition on ARI. Considering the statistical
results in Fig. 5, the average cloud liquid water content in CD and YB was significantly higher than
that in CQ under the influence of Patterns 2 and 5. Consequently, a more remarkable CRI
inhibition on the ARI would occur in the western and southern SCB compared to CQ, leading to a
relatively weaker ARI distribution in these regions. Without considering the CRI, the ARI in the
western and southern SCB would be much more pronounced than that in CQ. As for the
northwestern SCB (DZ), the ARI in DZ is lower than in the other three regions. When the CRI is not
considered, the ARI in DZ is higher than in CQ but lower than in CD and YB. This is because DZ has
lower aerosol concentrations compared to CD and YB (Fig. 7), but exhibits higher cloud cover than CQ
under Patterns 2 and 5 (Fig. 5).
Using the western SCB, which exhibited the highest pollution concentration, as an example, Fig.
9 illustrates the vertical diurnal variations in temperature and solar radiation caused by the ARI.
The results in Fig. 9–11 were derived from the simulation experiments in CD, as CD is one of the
most polluted cities with typical meteorological and geographical characteristics of the western SCB.
The ARI caused surface cooling in the morning and upper–air warming in the afternoon. As local
solar radiation increased from 8 am to 12 pm, the reduction in solar radiation caused by the ARI
also increased. Surface cooling reached its peak at approximately 10 am to 12 pm, and gradually
weakened in the afternoon. This diurnal variation might be attributed to the enhanced turbulence
during morning PBL evolution (Wang et al., 2018). Afternoon surface cooling was partly
compensated by the turbulent transport of warm air above the PBL. In addition, strong surface
cooling between 5 pm and 8 pm in the SCB, was possibly influenced by remarkable valley wind
circulations forced by the Qinghai–Tibet Plateau adjacent to the western SCB (Lu et al., 2022).
The evening cooling of the plateau induced strong mountain winds, promoting surface cooling,
while the upper–layer warming mainly occured around 1–1.5 km in the afternoon. In general, the
ARI reduces solar radiation, causing surface cooling and upper air warming, thereby regulating the
vertical atmospheric thermal structure, suppressing convection, and consequently decreasing PBL
heights (Fig. 10).

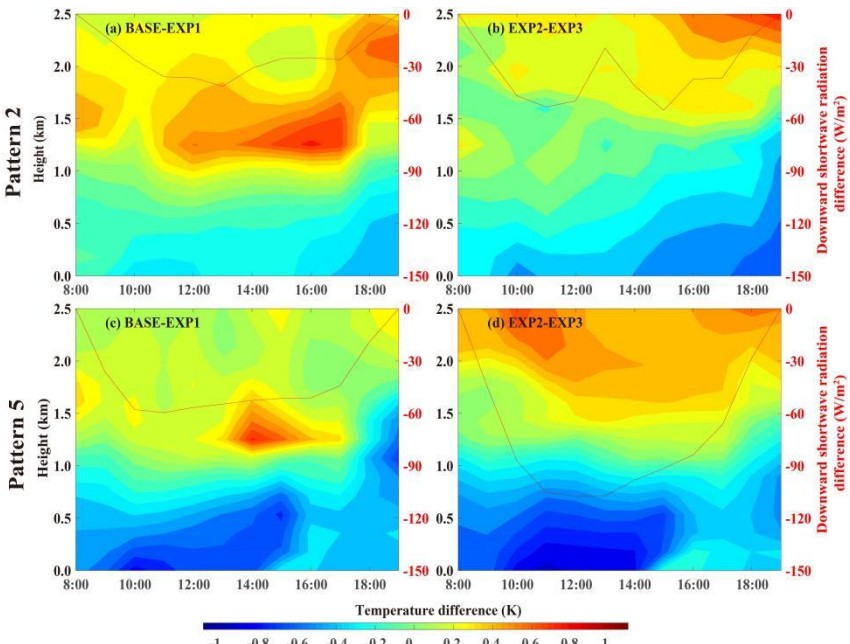

**Figure 9**. Diurnal variations of vertical temperature perturbations and downward solar radiation under influences of Pattern 2 and Pattern 5 induced by (a) (c) ARI and (b) (d) ARI without CRI inhibition.


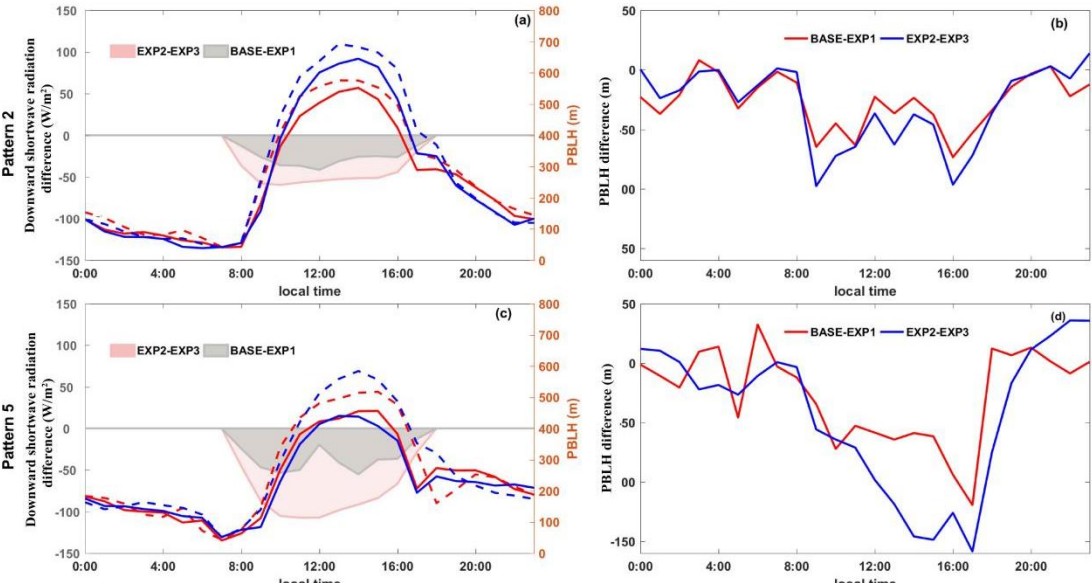

**Figure 10**. Diurnal variations of (a) (c) boundary layer height(lines) and downward solar radiation(shading), and (b)(d)the perturbations of boundary layer height induced by ARI and ARI without CRI inhibition, under Pattern 2 and Pattern 5 synoptic forcing respectively.


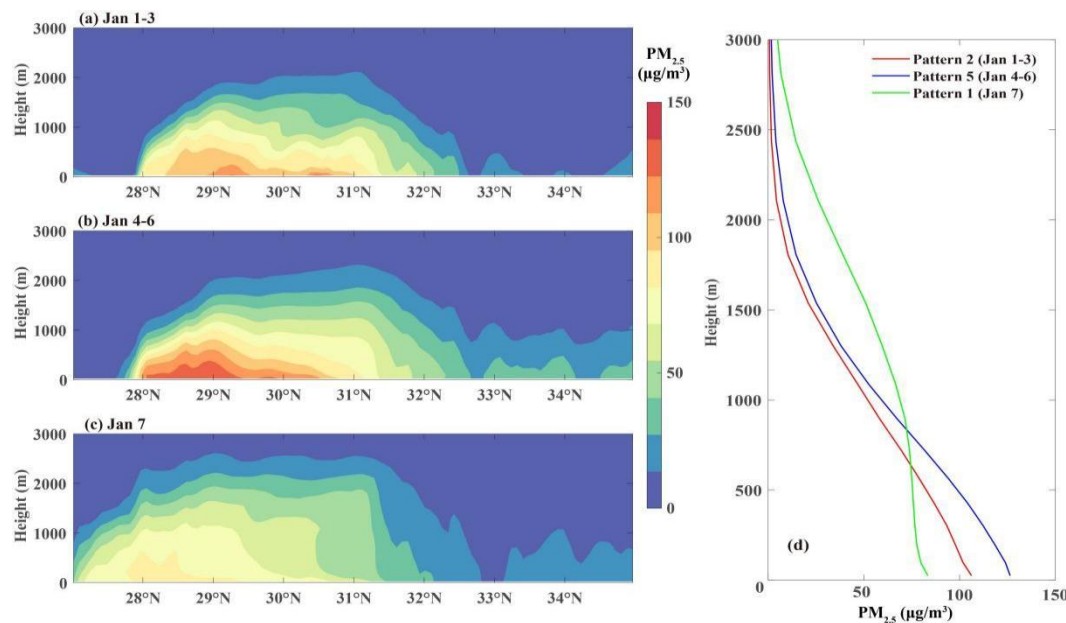

**Figure 11**. Meridional vertical distribution of averaged PM$_{2.5}$ between 104°E and 105°E under (a) Pattern 2 and (b) Pattern 5, and (c) average profiles of PM$_{2.5}$ within 28°N and 31°N.

Synoptic patterns play a role in the interaction between the ARI and PBL (Wang et al., 2018; Miao et al., 2020). Zonal average of PM$_{2.5}$ concentration between 104°E and 105°E was conducted, and the meridional vertical distribution of PM$_{2.5}$ between 27°N and 35°N was illustrated in Fig. 11a-b. Fig. 11(c) provides an average of PM$_{2.5}$ concentration within 28°N and 31°N, showing the vertical distribution profiles under Pattern 2 and 5. Due to the inhibition of "warm lid" above the SCB, the vertical exchange was not prominent under both Pattern 2 and 5, and PM$_{2.5}$ was more concentrated at the middle and lower levels. The PM$_{2.5}$ concentration under Pattern 5 was higher than Pattern 2 throughout the atmospheric column, indicating stronger aerosol radiative forcing and a more significant impact on the boundary layer structure under Pattern 5. During January 4–6, the surface cooling reached 1 K, with cooling layers higher than those observed on January 1–3. The differences in thermal structure modulations contributed to a lower diurnal PBLH in Pattern 5 than in Pattern 2 (Figs. 10a and c), indicating that Pattern 5 was more conducive to ARI. Based on the simulation experiments, this study further discussed the impact of synoptic forcing on the CRI inhibition of ARI. When the CRI was not considered, the solar radiation reduction at noon on January 4–6 by the ARI was nearly twice as high as when the CRI was considered. Correspondingly, surface cooling at noon was remarkably enhanced. In the evening, surface cooling occurred earlier and was stronger without the CRI (Fig. 9). The regulation of CRI on ARI was further reflected in changes in PBLH. Without the CRI, the diurnal PBLH increased significantly, with the PBLH decreased more with ARI without CRI inhibition. The PBLHs were decreased by the ARI during January 13–17 afternoon, reaching 2–3 times the decrease observed with CRI inhibition (Fig. 10). More significant CRI inhibition of ARI was revealed under Pattern 5 compared with that under Pattern 2, owing to the stronger ARI itself with higher aerosol

concentrations in Pattern 5 and the more apparent CRI inhibition with denser cloud liquid water
contents under the LT pattern (Fig. 5). Therefore, the intensity of CRI inhibition of ARI in the
SCB was altered by synoptic forcing, with stronger effects under the influence of LT.

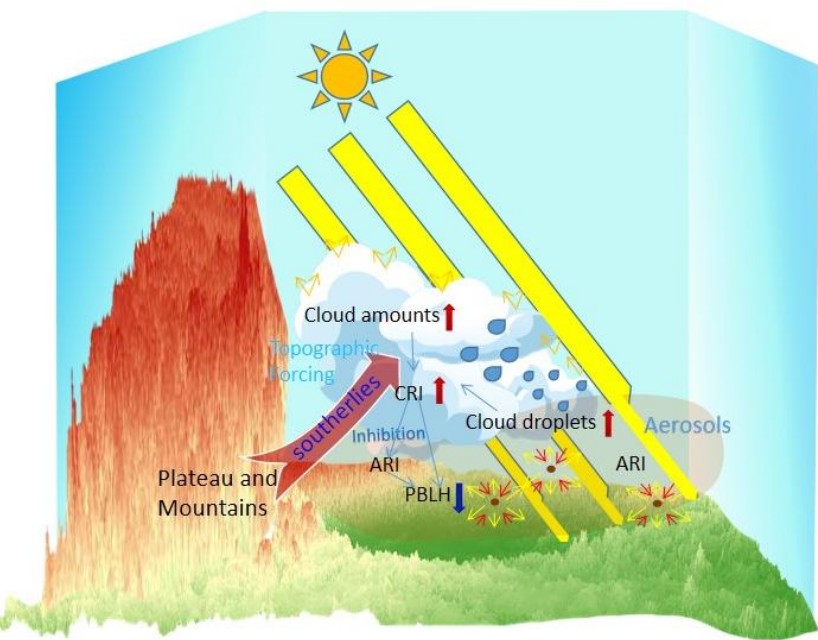

**Figure 12**. The aerosol radiative effect can be signigicantly inhibitted by cloud under influence of pollution synoptic patterns with dense cloud.


## 4 Conclusion


This study utilized synoptic classification and numerical simulation to gain insights in to the
combined effects of synoptic patterns and CRI inhibition on ARI and PBL structures in the wet and
cloudy SCB. Based on the long–term $PM_{2.5}$ observations and sounding data in the SCB, it was
found that large-scale synoptic circulations at 850 hPa played crucial roles in the variations of
$PM_{2.5}$ pollution. Synoptic classification was performed with the T-PCA method, which reveaed
that Pattern 2 and 5 characterized with low pressure system and southerly airflow on 850 hPa were
key synoptic patterns for onset and accumulation of $PM_{2.5}$, while Pattern 1 controlled by the
northerly airflow represented a clean pattern associated with significant decrease in $PM_{2.5}$.
Moreover, it was indicated that Pattern 2 and 5 exhibited denser cloud liquid water content and
thus stronger compared to other patterns. Among these patterns, Pattern 5 exhibited the highest
cloud liquid water content and CRI. This could be attributed to the robust southerly airflow
induced by the dense isobaric lines, which brought warm and humid air masses into the region.
To illustrate the interactions among cloud, aerosol and PBL under pollution synoptic patterns, a
pollution episode occurred from January 1 to 7, 2017, was simulated with using WRF-Chem. The

simulation results showed that ARI remarkably reduced solar radiation was during the two pollution patterns. This reduction led to surface cooling in the morning and upper–air warming in the afternoon. Additionally, the enhanced evening surface cooling was impacted by the mountain–valley wind circulations forced by the plateau–basin topography of the SCB. This modulation in the vertical thermal structure by the ARI would then suppress the development of the PBL, favoring pollution outbreaks (Fig. 12). Furthermore, parallel simulation experiments indicated that CRI impacted stratification stability and modulated the vertical thermal structure by inhibiting ARI (Fig. 12). Regarding the spatial distribution, a stronger ARI appeared in Chongqing, despite lower $PM_{2.5}$ concentrations compared to the western and southern SCB. This was due to the lower cloud liquid water content and weaker CRI inhibition of ARI in Chongqing. When CRI inhibition was not considered, the ARI in the western and southern SCB was significantly stronger than that in Chongqing. In addition, under Pattern 5, the reduction in solar radiation and PBLH during the daytime due to ARI could be more than doubled when the CRI influence was neglected. This was primarily due to higher aerosol concentrations and cloud liquid water contents associated with a low trough in Pattern 5. This study provided insights into the interaction among aerosols, clouds, and PBL under different synoptic patterns, considering the complex terrain and foggy/cloudy climate of the SCB. The findings highlighted the significant role of CRI inhibition on ARI during wet and cloudy conditions, shedding light in the multi-scale atmospheric physical processes in the SCB.

**Author contributions.** HL and MX had the original idea for the study, designed the experiments, conducted the numerical simulation and prepared the initial draft manuscript. BL, YZ and KZ collected the data. TW and BZ helped perform the analysis with constructive discussions, reviewed and edited the manuscript. HL, MX, TW and BL acquired financial support for the project leading to this publication. SL and ML reviewed the manuscript.

**Competing Interest:** The authors declare no conflict of interest.

**Data Available Statement**

The ERA5 pressure layer and single layer data can be respectively downloaded from https://cds.climate.copernicus.eu/cdsapp#!/dataset/reanalysis-era5-pressure-levels and https://cds.climate.copernicus.eu/cdsapp#!/dataset/reanalysis-era5-single-levels?tab=form. The NCEP FNL data are available at https://rda.ucar.edu/datasets/ds083.2/. The MEIC data can be accessed in Zheng et al (2018) at https://doi.org/10.5194/acp-18-14095-2018. Air quality and meteorological monitoring data can be acquired from https://doi.org/10.7910/DVN/USX59F.

**Financial support:** This work was supported by the National Natural Science Foundation of China (42205186, 42275102), the Chongqing Natural Science Foundation (cstc2021jcyj-msxmX1007), Special Science and Technology Innovation Program for Carbon Peak and Carbon Neutralization of Jiangsu Province (BE2022612), the key technology research and development of Chongqing

Meteorological Bureau (YWJSGG-202215; YWJSGG-202303) and  the research start-up fund for the
talented person recruitment of Nanjing Normal University (184080H201B57).

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

aerosol–radiation interactions on the effectiveness of emission control measures, Environmental
Research Letters 14, 024002.