# Peer review of "Impacts of synoptic forcing and cloud inhibition on aerosol radiative effect and boundary layer structure during winter pollution in Sichuan Basin, China"

_EGUsphere, 2023_

## Author Comment (AC1)

Dear Editors and Reviewers:

Thank you very much for your careful review and constructive suggestions with regard to our manuscript "Impacts of synoptic forcing and cloud inhibition on aerosol radiative effect and boundary layer structure during winter pollution in Sichuan Basin, China" (Manuscript Number: EGUSPHERE-2023-1806). Those comments are valuable and helpful for revising and improving our paper. We have studied these comments carefully and made changes in the manuscript according the reviewers' comments. The responses to the reviewer' comments are listed as follows, and the main corrections are marked with blue in the PDF file named "revised manuscript egusphere-2023-1806.pdf".

This study using long-term observation data from 18 air quality monitoring sites, combined with sounding data from four station in the Sichuan Basin, to explore the influence of synoptic forcing on the interactions between clouds, aerosols, and the PBL during wintertime. Further, the impacts of synoptic forcing and inhibition of cloud radiation interaction (CRI) on aerosol radiation interaction (ARI) with the PBL are discussed during an typical particle pollution episode. Some interesting results were present, but more in-depth analysis need to be provided. Here are some issues that need to be addressed for further improving this work.

Response: We are appreciated with your valuable and constructive comments, and have carefully considered these issues to improve this research.

1. Line 18-21, as six synoptic patterns were resolved and mentioned, a summary of these synoptic patterns should be provided.

Response: Thanks for pointing this out. We have incorporated a summary of these synoptic patterns in the abstract as: "The dominant 850 hPa synoptic patterns of winter SCB were classified into six patterns using T–model principal component analysis: (1) strong high pressure in the north, (2) east high west low (EHWL) pressure, (3) weak high pressure in the north, (4) weak ridge of high pressure after the trough, (5) low trough (LT), and (6) strong high pressure. ". Please see Line 20-22 of the revised manuscript.

2. Line 21-23, cloud liquid content is mentioned here, is this data obtained from the observation or from the model simulation? In addition, the processes of converted aerosols into fog/cloud

drops that contributed to the high cloud liquid content is subjective, the related discussion was not found in the main text.

Response: We are thankful for your comment and feel sorry for the confusion caused by these issues.

The cloud liquid content data is obtained from the ERA5 reanalysis data, as previous studies have demonstrated the reliability of ERA5 data in estimating cloud properties, including the cloud liquid content (Yao et al., 2019; Nandan et al., 2022; Ojo et al., 2023).

As for the statement regarding the processes of aerosol conversion into fog/cloud drops contributing to the high cloud liquid content, it is a speculation by authors based on previous researches (Twomey, 1977; Lohmann and Feichter, 2005; Boutle et al., 2018), but not fully discussed in the main text. For the accuracy and rigor of this study, we have removed the statement. Please see Line 182-186 of the revised manuscript.

3. Line 42-44, "Understanding .... is crucial for ...understanding... "it is not clear here, rewrite this sentence.

Response: We agree with the comment, and have rewritten this sentence in the revised manuscript as the following: "Studying interactions among cloud, aerosol and radiation from an air quality perspective is crucial for a scientific understanding of relationship between weather and pollution.". Please see Line 52-53 of the revised manuscript.

4. Line 45-47, what about the role of secondary aerosol formation?

Response: Thank you for pointing this out. In addition to the emission of primary aerosols, secondary aerosol formation also plays significant role in comprehending the complete picture of air pollution. Meteorological conditions not only influence the formation of secondary aerosols, but also govern the transportation and distribution of both primary and secondary aerosols, and thereby impact regional and long-range air pollution. We have rephrased the sentences in Line 54-58 of the revised manuscript.

5. Line 68-70, it is better to indicate the specific areas to which the conclusion applies.

Response: Thanks for the comment. We have rephrased it as: "This positive feedback between

unfavorable PBL meteorology and increasing aerosols was found to be responsible for the majority of the increase in $PM_{2.5}$ concentrations during cumulative stages in various regions of eastern China affected by aerosol pollution, including the North China Plain, the Guanzhong Plain, the Yangtze River Delta, the Two Lakes Basin, the Pearl River Delta and the Northeast China Plain. But in the Sichuan Bain, the feedback is weak due to the suppression of the cloudy mid-upper layer (Zhong et al., 2018; Zhong et al., 2019).". Please see Line 84-90 of the revised manuscript.

6. Line 88-91, The motivation of the present study was not clearly present, as the topic mentioned here had been reported previously for SCB region, like in Zhong et al., 2019.

Response: We are appreciated with the comment and feel sorry for not clarifying the motivation of this study clearly. In the revised version, we have added the description of the motivation of the study as: "Many studies have emphasized the importance of the interactions between cloud, aerosols and radiation in air pollution processes (Wang et al., 2018; Hu et al., 2021). High pollutant emissions, combined with the prevalence of cloudy and foggy weather, make these interactions in the SCB even more complex than those in other regions. The aerosol radiation interactions (ARI) can be inhibited by clouds in cities like Chengdu (Zhong et al., 2019). However, there is a lack of in-depth quantitative discussions regarding the aspects in the SCB. On one hand, the complex terrain in the SCB leads to differences in the meteorological conditions between them (Ning et al., 2018; Lu et al., 2022). For example, Chengdu is a typical basin city while Chongqing is a mountain city located on the basin slope, so they have markedly different climatic conditions. It remains to be elucidated whether these conditions will result in spatial disparities in cloud inhibition on the aerosol radiation interactions (ARIs). On the other hand, synoptic forcing, as the primary driver of meteorological variations, undoubtedly play an unneglectable role in shaping cloud cover and boundary layer structures (Miao et al., 2020; Wang et al., 2022; Painemal et al., 2023). The discrepancies in cloud inhibition on ARI under different synoptic patterns also need to be revealed. Addressing these issues is crucial for understanding the persistent pollution processes and the intricate interactions between weather and pollution in the SCB. It holds important implications for the effective management of pollution processes in cloudy and foggy weather.". Please see Line 108-124 of the revised manuscript.

7. Line 108-110, brief description about the difference in the air pollution and meteorological conditions for the selected four sounding station should be provided.

Response: Thank you for the suggestion. We have incorporated a brief description in the revised version as: "The SCB has four sounding stations Wenjiang (CD), YB, DZ, and Shapingba (CQ), which are situated in the western, southern, northwestern, and eastern regions of the basin, respectively (Fig. 1b), and represent different pollution and meteorological conditions in different regions within the SCB. In all, the air pollution over the SCB exhibits a gradual decrease from southwest to northeast. Statistical analysis indicates that the western and the southern basin experience the most severe pollution. The western basin shows the highest pollution proportion, while the southern basin exhibits the highest occurrence of heavy pollution. In the northeastern basin, specifically in Dazhou, heavy pollution is more likely to occur during winter, which verifies it to be the third highest pollution zone outside the western and southern basin. This makes the spatial distribution during winter differs from the overall annual pollution pattern in the SCB (Lu et al., 2022; Qi et al., 2022). Regarding meteorological conditions, research reveals that DZ has the lowest ventilation coefficient during winter, while CQ has the highest. The SCB experiences frequent temperature inversions, with CD having a higher occurrence of inversions compared to the other three cities. CD also exhibits the strongest inversion intensity and is prone to multi-layer inversions. On the other hand, YB and CQ have greater inversion thickness, while CD has the smallest inversion thickness (Feng et al., 2020)." Please see Line 146-161 of the revised manuscript.

8. Line 171-172, "other cities in the SCB also experienced pollution episodes and relevant physical processes". Do the authors mean other cities in the SCB showed consistent or similar pollution episodes and relevant physical processes? Please clarify it and provide the data to support it.

Response: Thank you for this rigorous comment. We have provided Figure S1 to show the time series of daily mean PM$_{2.5}$ and potential temperature derived from the ERA5 data of January 2017 for other fourteen cities in the SCB. The other cities have experienced the same pollution episodes and relevant physical processes, except for GY (Figure S1). GY is located in the northern edge of the SCB, bordering Shaanxi and Gansu Provinces. The proportion of heavy PM$_{2.5}$ pollution in GY

is the lowest in the basin, but the proportion of $PM_{10}$ pollution is higher than other cities of SCB (Lu et al., 2022). Due to the lower $PM_{2.5}$ concentration, the two pollution processes in January 2017 in GY were not as significant as in other cities within the basin. However, the warming of upper air coincided with $PM_{2.5}$ increase could still be observed. Please see Figure S1 in supplement and Line 254-259 of the revised manuscript.

9. Line 182, why high wind speed above 1500m would contribute to the rapid increase in $PM_{2.5}$?

Response: We are thankful for the comment. The issue can be explained by our previous study (Lu et al., 2022). We have added the corresponding explanation in the revised version as: "The previous study has found that winter heavy pollution processes in the SCB are usually associated with abnormal warming above the 850 hPa (Lu et al., 2022). The warming is induced by strong southerly airflow above the basin. The southerly airflow in winter over the SCB originates from the Yunnan-Guizhou Plateau or the Indian Peninsula, characterized with high temperature, dryness, and high wind speed. The strong southerly airflow forms a "warm lid" over the basin, suppressing the vertical exchange of pollutants within the basin. As a result, pollutants accumulate rapidly, which may explain the phenomenon of rapid $PM_{2.5}$ growth accompanied by warming, dryness, and strong winds above 1500 m.". Please see Line 272-279 of the revised manuscript.

10. Line 223-225, why pattern 4 was identified as synoptic pollution patterns? Any statistic results support it? In addition, about Pattern 6 that identified as the "clean pattern", the pollution occurrence frequency of which was much higher for the cities located in the eastern part of the SCB than the other parts (Figure 5a), explain it.

Response: Thanks for the valuable comment.

The reason why Pattern 4 was identified as a synoptic pollution pattern has been given in Line 306-309 of the revised manuscript. The reasons for Pattern 2 and 5 were also added. Some statistic results have been illustrated as: "Patterns 2, 4, and 5 exhibited higher frequencies of pollution occurrence ($PM_{2.5}$ daily concentration $\geq$ 75 µg/m$^3$) according to statistical results from 18 cities in the SCB during the 2015–2021 winters (Fig. 4a). " and "The days under Patterns 2, 4, and 5 exhibited higher average daily $PM_{2.5}$ concentrations. The average values under these three synoptic patterns were 99.19, 103.43 and 111.97 µg/m$^3$ for CD, 95.44, 87.98 and 94.26 µg/m$^3$ for

YB, 79.14, 83.96 and 74.77 μg/m$^3$ for CQ, and 91.02, 104.64 and 91.51 μg/m$^3$ for DZ, respectively. ".

The average PM$_{2.5}$ concentrations under Pattern 6 were lower in all cities of SCB than other three pollution patterns (Fig. 4a). Besides, the day to day PM$_{2.5}$ variations under Pattern 6 exhibited negative growth trend in the four representative cities (Fig. 4c). As a result, Pattern 6 was identified as the "clean pattern". For Pattern 6, strongest northerly airflow affected the basin. The eastern part of the basin consists of parallel ridges and valleys, which reduces wind speed. The stronger the wind is, the more obvious the reduction of wind by terrain is. In contrast, the western part is relatively flat, which can result in higher surface wind speeds. The difference in wind impacted by terrain led to a weaker pollution removal effect in the eastern region, thus contributing to a higher proportion of pollution days under Pattern 6. Besides, differences in precipitation rates between eastern cities and other regions were not significant (the proportion of rainfall with a daily accumulated precipitation exceeding 10 mm in CD, CQ, YB and DZ under Pattern 6 were all less than 3%), which might not the main reasons why eastern cities in the SCB experience higher pollution frequency. Please see Line 313-326 of the revised manuscript.

11. Line 243-236, Better to provide the precipitation information in Figure 6 to support this conclusion.

Response: Thanks for the suggestion. We have incorporated the daily accumulated precipitation in the figure. When the northerly airflow dominates the SCB (Pattern 1, 3 and 6), it is sometimes accompanied by weak precipitation that results to wet deposition and the removal of pollutants. Please see Figure S3 of the revised manuscript.

12. Line 247-248, it should be Pattern 6 with the highest PBL (1500m) as showed in Fig.7.

Response: We apologize for the error and have corrected it as: "The PBLH under Patterns 2 and 5 was approximately 900–1000 m, lower than that under the influence of clean synoptic Pattern 6 at 1500 m or Pattern 1 and 3 at 1200–1300 m (Fig. 5).". Please see Line 366-367 of the revised manuscript.

13. Line 271-274, If the low temporal resolution of sounding data made it not suitable to be adopted to compare, why the authors mentioned the good consistency of PBL in ERA5 with

sounding data in Guo et al., (2016)? Discussion about the comparison of PBL from the sounding data and the ERA5 is conflict here. At least, the PBL from sounding data and the comparison with the ERA5 and model results should be provided during the simulation period.

Response: Thanks for the comment. In Figure 9(a)-(d), the PBLH calculated based on the sounding data has been incorporated to compare with the ERA5 and model results during the simulation period. We calculated the PBLH with sounding data using the bulk Richardson number with a threshold value of 0.25, because the ERA5 and YSU boundary parameterization scheme also calculate the PBLH with the same method and the threshold (Hong et al., 2016; ECMWF, 2017). Sounding data are commonly regarded as reliable vertical observation records. PBLH calculated based on sounding data can be used as the true values to compare with other data for long-term validation (Guo et al., 2016). However, for short-term studies, due to limited availability of sounding data at only 00:00 and 12:00 UTC, the ERA5 data were also incorporated for the model evaluation of PBLH in this study (Fig. 6 and Fig.S5). The simulation PBLH showed a consistent trend with those calculated from ERA5 and sounding data. Please see Figure 6 and S5, Line 167-177 and Line 405-410 of the revised manuscript.

14. Line 341-343, As showed in Figure 13 and 15, the solar radiation and PBL induced by ARI was more distinct in Pattern 5 than in Pattern 2, data, and the authors attributed it to the denser aerosol in Pattern 5. Here, the denser aerosol is deduced from the higher $PM_{2.5}$ concentration from ground observation? If so, the analysis of the vertical profile of aerosol would be better to show the difference of aerosols' impacts on PBL development in Pattern 2 and Pattern5.

Response: We appreciate your valuable suggestion. Zonal average of $PM_{2.5}$ concentration between 104°E and 105°E was conducted, and the meridional vertical distribution of $PM_{2.5}$ between 27°N and 35°N was illustrated in Figure 11(a)-(b) in revised manuscript. Figure 11(c) provides an average of $PM_{2.5}$ concentration within 28°N and 31°N, showing the vertical distribution profiles under Pattern 2 and 5. Due to the inhibition of "warm lid" above the SCB, the vertical exchange was not prominent under both Pattern 2 and 5, and $PM_{2.5}$ was more concentrated at the middle and lower levels. The $PM_{2.5}$ concentration under Pattern 5 was higher than Pattern 2 throughout the atmospheric column, indicating stronger aerosol radiative forcing and a more significant impact on the boundary layer structure under Pattern 5. Please see Figure 11 and Line

486-493 in the revised manuscript.

---

## Author Comment (AC2)

Dear Editors and Reviewers:

Thank you very much for your careful review and constructive suggestions with regard to our manuscript "Impacts of synoptic forcing and cloud inhibition on aerosol radiative effect and boundary layer structure during winter pollution in Sichuan Basin, China" (Manuscript Number: EGUSPHERE-2023-1806). Those comments are valuable and helpful for revising and improving our paper. We have studied these comments carefully and made changes in the manuscript according the reviewers' comments. The responses to the reviewer' comments are listed as follows, and the main corrections are marked with blue in the PDF file named "revised manuscript egusphere-2023-1806.pdf".

**General Comments:**

The manuscript entitled "Impacts of synoptic forcing and cloud inhibition on aerosol radiative effect and boundary layer structure during winter pollution in Sichuan Basin, China" classifies the synoptic patterns influencing the SCB based on long–term data. Using WRF–CHEM simulation experiments, the impacts of synoptic forcing and inhibition of cloud radiation interaction (CRI) on ARI with the PBL in the SCB is analyzed. However, the authors are expected to carry out the following revisions as suggested below which may be helpful for the modification. In addition, the writing should be polished.

Response: We appreciate the reviewer for the valuable and constructive comments of our manuscript. We have carefully revised our manuscript based on the following comments and polished the writing throughout the whole manuscript.

**Specific Comments:**

1. The title "Impacts of synoptic forcing and cloud inhibition on aerosol radiative effect and boundary layer structure during winter pollution in Sichuan Basin, China" is conf Synoptic forcing refers to synergistic forcing of what and what? Winter pollution means the pollution of air or water or soil?

Response: Thanks for this rigorous comment. Synoptic forcing refers to synergistic forcing of synoptic atmospheric circulation patterns, and winter pollution here means winter air pollution. To make the expression more accurate, we have modified the title to be "Impacts of atmospheric circulation patterns and cloud inhibition on aerosol radiative effect and boundary layer structure

during winter air pollution in Sichuan Basin, China". Please see the title in the revised manuscript.

2. This author should be more careful. Extra spaces in the author's byline.

Response: We feel sorry for this formatting error and have made modification in the author's byline.

3. Lines 28-29: Sentence "Numerical simulation experiments using WRF-Chem showed afternoon upper–level heating and morning surface cooling forced by ..."should be improved.

Response: We are sorry for not expressing it clearly and have improved the sentence as "The results of numerical simulation experiments utilizing WRF-Chem indicated that there was a upper-level heating during afternoon and surface cooling in the morning forced by the aerosol radiation interaction (ARI) under the EHWL and LT patterns. Additionally, strong surface cooling in the evening influenced by valley winds can be found.". Please see Line 30-33 of the revised manuscript.

4. Lines 33: "thinner cloud liquid content" is confused? You mean low content? Improve it.

Response: Thank you for the careful comment. We have modified it as "lower cloud liquid content" in Line 36 of the revised manuscript.

5. Key words: Capitalize the first letter of each phrase.

Response: Sorry for the clerical errors. In the revised manuscript, we have capitalized the first letter of each phrase in Line 43-44.

6. Lines 33, 109, 179: Extra spaces in the

Response: Thanks for this careful comment. We have carefully checked the manuscript and deleted the extra spaces in the revised manuscript.

7. What are the boundary conditions, solver setup and governing equations.

Response: We are appreciated with the comment.

The initial and boundary conditions for the numerical model were obtained from the NCEP FNL reanalysis data with a horizontal resolution of $1° \times 1°$ and 6 h time interval. For

chemical process simulations, anthropogenic emissions were sourced from the Multiresolution Emission Inventory for China (MEIC) in 2016, featuring a grid resolution of 0.25° × 0.25°. To address the empirically overestimated $PM_{2.5}$ emissions by the MEIC in the SCB (Zhan et al., 2023), the ensemble square root Kalman filter were implemented on the $PM_{2.5}$ emission during simulation (Wu et al., 2018; Lu et al., 2021). Biogenic emissions were calculated online using the Guenther scheme (Guenther et al., 2006). The corresponding description could be found in Line 209-218 of the manuscript.

The main solver setup for WRF-Chem is presented in Table 1 in the manuscript as below:

Table 1 The main options of WRF–Chem

| Items | Contents |
|---|---|
| Domains (x, y) | (155, 110), (184, 160), (320, 250) |
| Grid spacing (km) | 27, 9, 3 |
| Center | (29.1° N, 106.2° E) |
| Time step (s) | 60 |
| Microphysics | WRF Single–Moment 5 class (WSM5) scheme |
| Longwave radiation | RRTMG scheme (Iacono et al., 2008) |
| Shortwave radiation | RRTMG scheme (Iacono et al., 2008) |
| Planetary boundary layer | Younsei University scheme (Hong et al., 2006) |
| Land surface | United Noah land surface model (Tewari et al., 2004) |
| Cumulus parameterization | Grell–Freitas ensemble scheme (Grell et al., 2013) |
| Advection | fifth- and third-order differencing for horizontal and vertical advection respectively |
| Photolysis scheme | Fast-J photolysis (Fast et al., 2006) |
| Gas–phase chemistry | RADM2 (Stockwell et al., 1990) |
| Aerosol module | MADE/SORGAM (Schell et al., 2001) |

The Advanced Research WRF (ARW) dynamics solver integrates the compressible, nonhydrostatic Euler equations, for example, the momentum equation, the continuity equation, the thermodynamic equation, the moisture equation and the ideal-gas equation of state. The details can be found in Skamarock et al. (2008).

We have added related descriptions in the revised manuscript. Please see Line 204-207 of the revised manuscript.

8. "The baseline experiment (BASE) considered both CRI and ARI, while the three sensitivity

experiments excluded ARI or CRI. Experiment 1 (EXP1) did not include ARI, Experiment 2 (EXP2) did not consider CRI, and Experiment 3 (EXP3) did not consider ARI when CRI was not excluded. The differences between BASE and EXP1 represented the disturbances caused by ARI, while EXP2 and EXP3 represented the influences of ARI without CRI inhibition.". Description on the case setup is a bit unclear. It is suggested to represent it in a table that illustrates the differences between experiments.

Response: We are appreciated with this suggestion, and have followed it by adding Table 2 in the revised manuscript.

9. "Figure 2. Time series of $PM_{2.5}$ and potential temperature derived from the sounding data during 2015-2021 winter months. The $PM_{2.5}$ pollution episodes are marked with black dotted " (1) The legend description should give the specific data time resolution in this graph. hourly or daily or monthly? (2) Try to give as much complete information as possible in the diagrams and don't use unofficial abbreviations, e.g. potential temperature abbreviations here.

Response: Thanks for the comments regarding Figure 2. In the revised manuscript, (1) The figure captions of Figure 2 and 3 have been modified with explicit time resolution to be "Time series of daily mean ..."; (2) The title of colorbar has included the full name of "Potential temperature" to avoid the use of unofficial abbreviations. Please see Figure 2 and Figure S2 in the revised manuscript and supplement.

10. Figures 4, 10, 11, 12. Mark the Sichuan Basin in this figure.

Response: We are appreciated with the suggestion and have marked the Sichuan Basin outline with an altitude contour of 750 m terrain height in the corresponding figures. To streamline the manuscript, we have moved some figures to the supplement and merged Figure 10 and 11. As a result, the orders of the figures in the revised manuscript have been changed. Please see Figure 3, 7 and 8 in the revised manuscript.

11. Figure 6: Relationship of $PM_{2.5}$ concentrations and the day to day 850 hPa synoptic patterns is not pictured clearly and should be improved.

Response: Thanks for the suggestion. We have added some arrows in Figure 6 to make the

relationship of daily mean $PM_{2.5}$ and day to day 850 hPa synoptic patterns clearer. It is obvious that pollution episodes always begin with Pattern 2 and ended with Pattern 1, suggesting that Pattern 2 is the key synoptic pattern for pollution initiation. Besides, Figure 5 provides a statistically and quantitative clearer representation of the relationship between daily mean $PM_{2.5}$ and day to day 850 hPa synoptic patterns. Please see Figure S3 in the revised supplement.

12. Lines 308-309: Figure 8: The section on model validation is not sufficiently analyzed, especially the lack of quantification of the correlation between the model and the observations, the assessment of the consistency, and the explanation of the reasons for the large differences, such as wind speeds.

Response: Thank you for the valuable comment regarding the evaluation of model performance. We have provided the statistical metrics between the simulation and observation of $PM_{2.5}$ and the meteorological factors in the revised Figure 6 and S4. The MB of the simulated and observed $PM_{2.5}$ concentrations were -15.59, -13.42, 2.10 and -13.11 μg/m³, respectively, with NMB values of -4.12%, -4.22%, 6.01% and -0.68%, respectively, which are within the acceptable standards (NMB < ± 15%). The R of $PM_{2.5}$ are 78.91%, 57.23%, 61.15% and 62.86% for 4 representative cities, respectively. The statistical metrics for $PM_{2.5}$ are consistent with previous studies (Wang et al., 2020; Shu et al., 2021; Zhan et al., 2023), indicating that our model results for $PM_{2.5}$ are reasonable and acceptable. Regarding to the surface meteorological factors, low MB and high R for both temperature and dew point temperature suggested good simulation performances for these variables. However, the simulation results for wind speed were poor, which was expected under conditions of low wind and complex terrain. Due to the starting speed of the anemometer and high calm wind frequency in the SCB, the disturbances of observed wind speeds were usually smaller than simulation. This led to a significant deviation in the simulation results (Shu et al., 2021; Zhan et al., 2023). Additionally, it could be argued that unresolved topographic features introduce additional drag, beyond that generated by vegetation, which was not considered in the WRF model (Jimenez and Dudhia, 2021). These findings indicate that the simulation of $PM_{2.5}$ and meteorological factors is reasonable in the SCB; thus, the simulations can be used for subsequent analysis. Please see Figure 6 and S4 of the revised manuscript and supplement. The corresponding description can be found in Line 381-396 of the revised manuscript.

13.  Lines 312-313: "The simulation aligned well with the sounding observations, reflecting upper air warming and PBL humidification during the accumulation process of PM2.5 ..."is not appropriate. With regard to the comparison between simulated and observed vertical profiles, the results are not very good. Even the temperature, which is best simulated accurately, shows a maximum difference of up to 5 ℃, mainly within the boundary layer (below 1.5km), so is the choice of parameterization scheme open to question? Not to mention the relative humidity and wind speed, which are numerically very different, but the trending is okay.

Response: Thanks for indicating this. Regarding to this issue, we have checked the drawing code and found that we have not unified the color scale when drawing the vertical distribution figures, which caused significant differences in temperature values within the boundary layer in the figure. However, based on the variations of surface temperature (Figure 6 and S4), it can be seen that there are no so many differences in surface temperature. We have unified the color scale in the revised figure. In terms of the choice of parameterization schemes, the schemes employed in this study is the one used by the Chongqing Meteorological Bureau in the daily operational activities. The schemes have been obtained through multiple sets of control experiments and are considered suitable for the simulation in the SCB. Additionally, we have also attempted to conduct the simulation experiment with parameterization schemes proposed by previous studies relating to the SCB simulation. The figure below illustrates the comparative results among simulations with parameterization schemes of this study, Wang et al. (2020) and Zhan et al., (2023). It can be seen that these schemes do not yield better simulation results than our chosen schemes when compared with the sounding data. The SCB is characterized with cloudy and foggy conditions, which result in abundant water vapor and near 100% relative humidity above the nocturnal boundary layer. Models often underestimated the humidity above the boundary layer during night in the SCB (Shu et al., 2021). Furthermore, due to complex terrain and measurement bias of the anemometer for weak winds, the evaluation of simulation results for wind speed often exhibited certain deviations (Jimenez and Dudhia, 2021; Shu et al., 2021; Zhan et al., 2023). Overall, the simulation results can capture the meteorological variation trends. According to the simulation evaluation standards for the SCB in previous studies (Wang et al., 2020; Zhan et al., 2023), the simulation results is acceptable and can be used for subsequent analysis and discussion.

In the revised manuscript, we have updated Figure 6 and S5. The description of simulation results has also been refined to make it more consistent with the figures. Please see Figure 6 and S5, and Line 397-413 of the revised manuscript.

[Figure]

Figure 1 Comparison between observations and simulation results using different parameterization schemes in this study, Wang et al. (2020) and Zhan et al. (2023).

14. Figure 9. Regarding the boundary layer height, illustrate the determination of BLH from simulations. From what seen in Figure 9, I think your BLH is not consistent with the potential temperature profiles and wind profiles. And according to the literature below, BLH determination method is relative to boundary layer structure. 1016/j.atmosres.2020.105179

Response: We are appreciated with the comment and feel sorry for not clarifying clearly about the method and data to calculate the PBLH.

There are various methods to determine the PBLH, as Jiang et al. (2021) mentioned by the reviewer, who estimated the thermodynamic and material PBLH based on MWR with the parcel method and ceilometer, respectively. The dynamic PBLH can be calculated by the the bulk Richardson number ($Ri$) method or some other methods, like derived from the TKE profiles. The differences in methods, data or threshold values may yield quite different PBLH results (Seibert et al., 2000; Eresmaa et al., 2006; Jiang et al., 2021).

The hourly PBLH variations (red line) in Figure 9(a)-(d) are obtained from ERA5 data. In the

revised figure, PBLH calculated based on sounding data (green dots) is also included in Figure 6(h) and S5(a)-(d), despite its low temporal resolution and inability to capture the peak values of diurnal PBLH. The hourly PBLH variations (red line) in Figure 6(k) and S5(e)-(h) are derived from simulation results. Both the EAR5 and YSU schemes use the $Ri$ method to calculate the PBLH, with a threshold value of 0.25 (Hong et al., 2006; ECMWF, 2017). Therefore, the PBLH calculation based on sounding data also employs the $Ri$ method and the same threshold value.

Theoretically, the variation of PBLH should be consistent with the variation of potential temperature or wind speed. However, the PBLH calculated using the $Ri$ method may not perfectly align with the isentropes or wind speed contours, but just exhibits the similar trends. Furthermore, the potential temperature and wind speed in Figure 6 and S5 are obtained at 12-h intervals (00 and 12 UTC), with the same temporal resolution as the sounding data, which may not fully capture the diurnal peak and finer variations. But the PBLH derived from ERA5 and simulation results are provided at hourly intervals. Differences in temporal resolutions may also result in some discrepancies between variations in the PBLH and the potential temperature or wind speed.

We have added details of the method and data to calculate the PBLH in the revised manuscript. Please see Figure 6 and S5 of the revised manuscript and supplement. Corresponding description can be found in Line 167-177 of the revised manuscript.

15. Figure 10: The magnitude of the wind speed represented by the arrow vectors is not specified.

Response: We feel sorry for the neglect, and have added the specific magnitude of the wind speed in the legend of Figure 10. Please see Figure 7 of the revised manuscript.

16. Too many pictures in the article, Figures 10 and 11 are suggested to be merged.

Response: Thank you for the suggestion. We have merged Figure 10 and 11 to reduce the number of pictures and improve the overall flow and readability. Please see Figure 7 of the revised manuscript.

17. For all images, both in the picture and in the title, lack of space between brackets and words.

Response: Thanks for pointing this out. We have modified all figures and their titles concerning this issues. Please see figures and their titles in the revised manuscript.

18. Figure 15. This abstract figure is good. But the title "The Synergetic interactions of cloud, aerosol and radiation under the influence of cloudy pollution synoptic forcing." is a little confused. And the second word should not be capitalized.

Response: We are sorry for any confusion caused by the captain of Figure 15, and have rephrased the sentence. Please see captain for Figure 12 of the revised manuscript.

19. The article is overloaded with graphs, some graphs such as Figures 5 and 6 express similar results and meanings, and can make trade-offs. The focus of the article's diagrams is unclear; some of the diagrams could be placed in supplemental material.

Response: We are appreciated with the valuable comment and have adjusted with diagrams in the manuscript. Figure 3 and 6 have been moved into the supplement, while Figure 10 and 11 have been merged. Besides, Figure 8 and 9 have been moved to the supplement, and information in CD have been merged to be Figure 6 as an example. Please see figures in the revised manuscript and the supplement file.

---

## Author Response (AR2)

Dear Editors and Reviewers:

Thank you very much for your careful review and valuable suggestions with regard to our manuscript "Impacts of synoptic forcing and cloud inhibition on aerosol radiative effect and boundary layer structure during winter pollution in Sichuan Basin, China" (Manuscript Number: EGUSPHERE-2023-1806). Those comments are helpful for revising and improving our paper. We have studied these comments carefully and made modifications in the manuscript according the reviewers' comments. The responses to the reviewers' comments are listed as follows.

This study classified the synoptic patterns influencing the SCB and revealed the impacts of meteorological factors on $PM_{2.5}$ concentrations. Using WRF-CHEM simulations, the authors explored the influence of synoptic conditions and cloud radiation interaction (CRI) on aerosol radiation interaction (ARI) and PBL structures during a high $PM_{2.5}$ event. The findings contribute to the understanding of CRI, ARI, and the PBL interactions in regions with wet and cloudy weather. This paper is rightfully within the scope of ACP. However, certain sections require further clarification to enhance the paper's clarity. Please find my detailed comments below for your consideration.

Response: We are appreciated with your valuable comments, and have carefully considered the issues below to improve this research.

1. Line 90-91: What does the unit 'h' in cloud cover stand for?

Response: We are sorry for the clerical error, and have rephrased this sentence as: The mean annual relative humidity in the SCB is around 75%, with cloud fraction exceeding 80%, and an average of 1200 hours of sunshine per year. Please see Line 98-100 of the revised manuscript.

2. Line 136: Provide the full name of "ECMWF".

Response: Thanks for this careful comment. "ECMWF" is short for "European Centre for Medium-Range Weather Forecasts". We have added the full name when first mentioned it in Line 186 of the revised manuscript.

3. Line 139: What criteria were used to select these four representative cities?

Response: Thank you for this comment. We selected these four representative cities in the western,

southern, northwestern, and eastern regions of the basin, to capture diverse pollution and meteorological conditions within the SCB. These cities are chosen to represent the most polluted regions (Zhao et al., 2018; Lu et al., 2022), as well as typical basin and mountainous cities. Furthermore, there are only four sounding stations in the SCB available: Wenjiang (CD), YB, DZ, and Shapingba (CQ). These stations in the four cities can provide valuable vertical and surface meteorological observations, as well as pollution data, contributing comprehensive dataset used in this study. We have added the reason why we choose these four cities in Line 178-185 of the revised manuscript.

4. Line 169: Are chemical initial and boundary conditions considered in this study?

Response: Thanks for the comment. We used the default chemical initial and boundary conditions. However, we made specific adjustments to the anthropogenic emissions in this study. To address the empirically overestimated $PM_{2.5}$ emissions by the MEIC in the SCB (Zhan et al., 2023), the ensemble square root Kalman filter were implemented on the $PM_{2.5}$ emission during simulation (Wu et al., 2018; Lu et al., 2021). Additionally, the first 48 hours of the simulation were designated as the model spin-up period. As a result, the simulation results for $PM_{2.5}$ concentrations are acceptable in this study. Please see Line 223-226 and Line 245-247 in the revised manuscript for details.

5. Line 173: Change "the impact of CRI inhibition on ARI and ARI" to "the impact of CRI inhibition on ARI".

Response: We are thankful for your kind remind and have followed the suggestion in Line 234 of the revised manuscript.

6. Line 174-176: It is more logical to state that you utilized this MEIC emission inventory because it aligns closely with the study period.

Response: Thanks for the comment. We have rephrased this sentence in Line 236-239 of the revised manuscript to make it more logically.

7. Line 178: Why is the year 2017 the key year about current $PM_{2.5}$ pollution?

Response: We are sorry for not clarify it clearly. The Chinese government announced clean-air

action in the year of 2013, aiming to reduce $PM_{2.5}$ concentrations in the next 5 year. Specifically, the year of 2017 was identified as a key year for assessing $PM_{2.5}$ pollution in China, as significant practical actions were implemented during the period. We have added some more description about this in Line 236-238 of the revised manuscript.

8. Line 181-186: Please provide a clearer explanation of EXP3. Why do EXP2 and EXP3 represent the influences of ARI without CRI inhibition?

Response: We apologize for any confusion. In EXP3, both ARI and CRI were shut down, while in EXP2, only CRI was omitted. By comparing the results of EXP2 and EXP3, we can isolate and assess the specific influence of ARI without the presence of CRI inhibition. In order to clarify the experimental setup, we have added Table 2 in the revised manuscript to provide a clear overview of the four numerical simulation experiments.

9. Line 200-202: Are there any observations supporting this point?

Response: We are appreciated with this rigorous comment. We have added Figure S1 in the revised supplement, and some further analysis and discussion regarding this point are also provided. Please see Line 262-264 of the revised manuscript and Figure S1 in the supplement.

10. Line 207-209: What does "influencing all four cities" refer to?

Response: We feel sorry for the confused description and have rephrased the description as "During this month, two severe $PM_{2.5}$ pollution episodes occurred: one from January 1 to 7 and another from January 24 to 31 in 2017. These pollution episodes had a significant impact on air quality in all four cities. The highest daily $PM_{2.5}$ concentrations recorded during these episodes were 291.17 $\mu g/m^3$ in CD and 276.21 $\mu g/m^3$ in YB.". Please see Line 274-277 of the revised manuscript.

11. Figure 4: It would be better to plot the outline of the SCB in this figure to enhance readability.

Response: We appreciate for your kind suggestion and have added the outline of the SCB in the figures. Please see Figure 3, 7 and 8 in the revised manuscript.

12. Line 245: Ensure all data is presented with the same number of decimal places.

Response: Thanks for this careful comment. We have unified the number of decimal places in Line 317-320 of the revised manuscript.

13. Figure 5c: The unit should be "μg/m3".

Response: We feel sorry for the mistake and have made modification. Please see Figure 4 of the revised manuscript.

14. Line 267: Clarify which direction of airflow controls the upper-layer of the basin.

Response: Thanks for your kind remind. It should be "when the southerly airflow controlled the upper-layer of the basin". We have made modification in Line 355 of the revised manuscript.

15. Line 269-279: Support your explanation with relevant figures, data, or existing research results.

Response: Thank you for your comment. The issue can be explained by our previous study (Lu et al., 2022). This study indicated that a "warm lid" appeared when the southerly airflow controlled the SCB around 850 hPa, suppressing the vertical exchange of pollutants within the basin. Forced by the surrounding high mountains, pollutants can fully mixed and chemical reactions conduct through secondary circulation in the basin. When the northerly airflow began to dominate the SCB, the "warm lid" was disrupted, leading to dispersion of pollutants through vertical transport. The evolution of 850 hPa synoptic forcing and vertical meteorological conditions (Figure 2 and 6) aligns with the study of Lu et al (2022). Therefore, there are also similar pollution change mechanisms. Actually, Figure 12 and the relate descriptions also can support the similarities in mechanisms between the two studies. Please see the descriptions in Line 357-370 and Figure 12 of the revised manuscript.

16. Line 376: What do the other two regions represent? You mentioned four cities in your previous results.

Response: We are sorry for the neglect. The southern SCB (YB) and the western SCB (CD) share similarities in terms of topography, but quite different with the eastern SCB (CQ). A strong ARI was primarily observed in CQ, as well as in the western and southern SCB, despite CQ

experiencing lower pollutant concentrations compared to the other two regions (Figs. S4 and 7). Without considering the CRI, the ARI in the western and southern SCB would be much more pronounced than that in CQ. This is due to obviously higher cloud cover under Patterns 2 and 5 in CD and YB compared to CQ (Figure 5). As for the northwestern SCB (DZ), we have added some relate analysis. The ARI in DZ is lower than in the other three regions. When the CRI is not considered, the ARI in DZ is higher than in CQ but lower than in CD and YB. This is because DZ has lower aerosol concentrations compared to CD and YB, but under the influence of weather patterns 2 and 5, DZ exhibits higher cloud cover than CQ. Please see Line 469-482 of the revised manuscript for the modification.

17. Line 406-408: The difference in CRI between Pattern 2 and Pattern 5 may also contribute to varied surface cooling.

Response: Thanks for the valuable comment. We admit that the stronger surface cooling during Pattern 5 may also due to the differences in CRI. To provide a more visual representation of the comparison, we have added Figure 12. The figure shows the $PM_{2.5}$ concentration under Pattern 5 was higher than Pattern 2 throughout the atmospheric column, indicating stronger aerosol radiative forcing and a more significant impact on the boundary layer structure and surface cooling under Pattern 5. Please see Figure 12 and Line 509-511 of the revised manuscript.

Sincerely,
Authors